# Niobium-doped layered cathode material for high-power and low-temperature sodium-ion batteries

Qinhao Shi[1,7], Ruijuan Qi[2,7], Xiaochen Feng[1,7], Jing Wang[3,7], Yong Li[1], Zhenpeng Yao [4], Xuan Wang[1], Qianqian Li[1], Xionggang Lu[5], Jiujun Zhang [1,6] & Yufeng Zhao [1✉]

The application of sodium-based batteries in grid-scale energy storage requires electrode materials that facilitate fast and stable charge storage at various temperatures. However, this goal is not entirely achievable in the case of P2-type layered transition-metal oxides because of the sluggish kinetics and unfavorable electrode|electrolyte interphase formation. To circumvent these issues, we propose a P2-type $Na_{0.78}Ni_{0.31}Mn_{0.67}Nb_{0.02}O_2$ (P2-NaMNNb) cathode active material where the niobium doping enables reduction in the electronic band gap and ionic diffusion energy barrier while favoring the Na-ion mobility. Via physicochemical characterizations and theoretical calculations, we demonstrate that the niobium induces atomic scale surface reorganization, hindering metal dissolution from the cathode into the electrolyte. We also report the testing of the cathode material in coin cell configuration using Na metal or hard carbon as anode active materials and ether-based electrolyte solutions. Interestingly, the Na||P2-NaMNNb cell can be cycled up to 9.2 A g$^{-1}$ (50 C), showing a discharge capacity of approximately 65 mAh g$^{-1}$ at 25 °C. Furthermore, the Na||P2-NaMNNb cell can also be charged/discharged for 1800 cycles at 368 mA g$^{-1}$ and −40 °C, demonstrating a capacity retention of approximately 76% and a final discharge capacity of approximately 70 mAh g$^{-1}$.

[1] Institute for Sustainable Energy/College of Sciences, Shanghai University, Shanghai 200444, P. R. China. [2] Key Laboratory of Polar Materials and Devices (MOE) and Department of Electronics, East China Normal University, Shanghai 200062, P. R. China. [3] Key Laboratory of Applied Chemistry in Hebei Province, Yanshan University, Qinhuangdao 066004, P. R. China. [4] Center of Hydrogen Science, Shanghai Jiao Tong University, 800 Dongchuan Road, Shanghai 200240, P. R. China. [5] State Key Laboratory of Advanced Special Steel & Shanghai Key Laboratory of Advanced Ferrometallurgy & School of Materials Science and Engineering, Shanghai University, Shanghai 200444, P. R. China. [6] College of Materials Science and Engineering, Fuzhou University, Fuzhou 350108, P. R. China. [7] These authors contributed equally: Qinhao Shi, Ruijuan Qi, Xiaochen Feng, Jing Wang. ✉email: yufengzhao@shu.edu.cn

As an alternative to lithium-ion batteries, sodium-ion batteries (SIBs) have received significant attention for grid-scale energy storage due to the relatively high abundance of sodium sources. Nevertheless, the practical application of SIBs is hindered by the slow $Na^+$ transfer kinetics, especially at subzero temperatures. Layered transition-metal oxides ($Na_xTMO_2$, where TM represents a transition metal) are considered the most promising cathode families due to their low cost and high theoretical capacities[1–9]. Among the layered structures (mainly P2 and O3, according to the count of edge-sharing $TMO_6$ octahedra with oxygen piling in ABBA or ABCABC stacking[10] as displayed in Supplementary Fig. 1), $P2-Na_{2/3}Mn_{2/3}Ni_{1/3}O_2$ has been extensively investigated and demonstrated to have a notable specific capacity of 160 mAh $g^{-1}$ within 2.0–4.5 V based on $Ni^{2+}$/$Ni^{4+}$ redox couples[11–13]. Nevertheless, this material suffers from irreversible structural changes or phase transitions (e.g., P2-O2, P2-OP4) at certain desodiation states[14] and severe interfacial transition metal dissolutions[15–20], thus causing a wretched rate capability and poor capacity decay upon cycling.

Efforts have been devoted to stabilizing the crystal structure during charge/discharge and enabling fast $Na^+$ (de)intercalation. For instance, proper cation doping (e.g., $Li^+$ [14,21], $Cu^{2+}$ [22], $Mg^{2+}$, [23,24], $Zn^{2+}$ [22]) has been proven to be efficient in eliminating TM layer gliding and $Na^+$/vacancy ordering, thereby realizing a fast solid-solution reaction[25]. Surface modifications have been applied to improve the surface structural stability upon cycling[19,20,26] while increasing the Na content was reported to be capable of stabilizing the crystal structure at deep desodiation states[14,27]. For example, by increasing the sodium content to 0.85, and including with $Li^+$ substitution[28], a plateau-free P2-type cathode, $Na_{0.85}Li_{0.12}Ni_{0.22}Mn_{0.66}O_2$ (P2-NLNMO), was developed[14], which enables fast $Na^+$ mobility ($10^{-11}$ to $10^{-10}$ $cm^2$ $s^{-1}$) by realizing a completely solid-solution reaction over the whole experimental potential range. In these studies, a sufficient Na content and complete solid-solution reaction are considered critical to achieving high-rate and highly stable P2-type cathodes[14,27]. However, regardless of the achievements obtained at room temperature, such structures generally encounter a noticeable capacity loss at subzero temperatures[25,29,30].

In this study, we report the preparation, physicochemical and electrochemical characterizations of a Nb-doped P2-type cathode active material (i.e., $Na_{0.78}Ni_{0.31}Mn_{0.67}Nb_{0.02}O_2$ hereinafter P2-NaMNNb) capable of efficient battery cycling at low temperatures (LTs) such as −40 °C. Nb doping can simultaneously reduce the electronic band gap and ionic diffusion energy barrier, thus enabling fast electron and $Na^+$ transfer especially in the Na-deficient state. Moreover, a cation-mixed layer is formed on the surface of P2-NaMNNb along the direction of $Na^+$ diffusion channels at an atomic scale (which enables the formation of a thin but robust cathode-electrolyte interphase (CEI)), thus enhancing charge transfer and inhibiting transition metal dissolution upon cycling. It should be noted that, the Nb (0.02, stoichiometric value) substitution enables a relatively high Ni content (0.31, stoichiometric value) in high Na-content P2-type cathode material, thus maintaining the voltage plateaus of $Ni^{2+}$/$Ni^{3+}$/$Ni^{4+}$ at 4.15~3.3 V. Consequently, the P2-NaMNNb materials exhibit a capacity of 96.6 mAh $g^{-1}$ at 92 mA $g^{-1}$ with a working voltage of 3.4 V (voltage range: 2.4~4.15 V), and an appealing rate capacity (65.8 mAh $g^{-1}$ at 9.2 A $g^{-1}$ (50 C)). Interestingly, P2-NaMNNb presents almost unchanged diffusion coefficients from 25 to −40 °C ($\approx 10^{-9}$ $cm^2$ $s^{-1}$), which enables cycling of Na||P2-NaMNNb coin cells at 1.84 A $g^{-1}$ and −40 °C, showing a stable specific discharge capacity of approximately 63 mAh $g^{-1}$. For the same cell configuration, we also report 1800 cycles at 368 mA $g^{-1}$ and −40 °C with a capacity retention of approximately 76% and a final discharge capacity of approximately 70 mAh $g^{-1}$. P2-

NaMNNb was also tested in combination with a hard carbon (HC) anode active material. The HC||P2-NaMNNb coin cells demonstrate maximum specific energy and specific power of 202 Wh $kg^{-1}$ and 7.75 kW $kg^{-1}$ (values calculated using the total mass of the positive and negative electrode active materials), respectively, at 25 °C.

## Results

### Structural and chemical characterizations of P2-NaMNNb.
The crystal structure of the as-prepared sample was determined by powder X-ray diffraction (XRD) and Rietveld refinement. The refined results of P2-NaMNNb and P2-NaMN ($Na_{0.78}Ni_{0.32}Mn_{0.68}O_2$) fit well with the experimental data (Fig. 1a, b). All the diffraction patterns can be indexed into a hexagonal structure with the space group $P6_3/mmc$, whereby Mn, Ni, and Nb atoms are settled on the 2a site of the transition-metal layer. There are two kinds of prismatic sites in the Na layer, $Na_f$ and $Na_e$, sharing two faces or edges with the lower and upper octahedral $TMO_6$, respectively (Supplementary Fig. 1). As already reported in the literature[27], a preferred occupancy at $Na_e$ sites ($Na_e$: 0.543 mol, $Na_f$: 0.237 mol) is observed in P2-NaMNNb to achieve lower electrostatic repulsion between Na elements. The doping with Nb causes slight shifts of the (002) and (004) peaks to a lower angle (Supplementary Fig. 2), leading to an increase in the d-spacing according to the Bragg equation[31]. It is worth noting that both compounds showed a preferred orientation of XRD intensity for the (104), (100), and (110) planes, which might be attributed to the coprecipitating $(Mn_{0.67}Ni_{0.31})_xCO_3$ synthesis method applied in this work[32]. The elemental ratio of Na, Mn, Ni, and Nb in P2-NaMNNb was determined to be 0.78:0.67:0.31:0.02 by the inductively coupled plasma atomic emission spectrometry (ICP-AES) method (Supplementary Table 1). As illustrated in Fig. 1c, the refined crystal structure indicates that Nb doping can expand the distance between the TM layers from 0.376 to 0.389 nm and the Na-O bond from 0.251 to 0.256 nm, which endows $Na^+$ with enhanced de/intercalation capabilities (Supplementary Tables 2 and 3).

### Electron microscopy and theoretical characterizations of P2-NaMNNb.
Aberration-corrected high-angle annular dark-field scanning transmission electron microscopy (AC-HAADF-STEM) was carried out to intensively investigate the effect of Nb doping on the bulk and surface structure of P2-NaMNNb at the atomic scale (Fig. 2a, b, Supplementary Fig. 3). As the atomic images show in Fig. 2b, c, it is possible to observe grains projected along the [010] zone axis, with the large bright representing TM atom columns, but light elements, such as Na and O, are not detectable, which fits with the representatively simulated P2-layered atomic models using the $P6_3/mmc$ space group as shown by the inset. The perfect lattices (without dislocation or defects) illustrate the high crystalline quality of the obtained P2-NaMNNb. In addition, indexed from the fast Fourier transform (FFT) patterns (Supplementary Fig. 4a) of the HAADF image in Fig. 2b, the space distances of the lattice fringes for P2-NaMNNb are 0.558 nm and 0.249 nm corresponding to the (002) and (100) planes (enlarged view of the green rectangle of Fig. 2b), respectively (Fig. 2c). The c-axis corresponding to the (002) plane visibly varied when compared with that of P2-NaMN (Supplementary Fig. 4b), which agrees well with the results of XRD refinement. Moreover, as shown in Fig. 2d, it was found that the atom arrangement in the surface region (with a thickness of 3–5 nm) of P2-NaMNNb is different from that of the bulk, which indicates that Nb doping might induce chemical changes in the surface layer of the material. As reported in the literature, such a surface

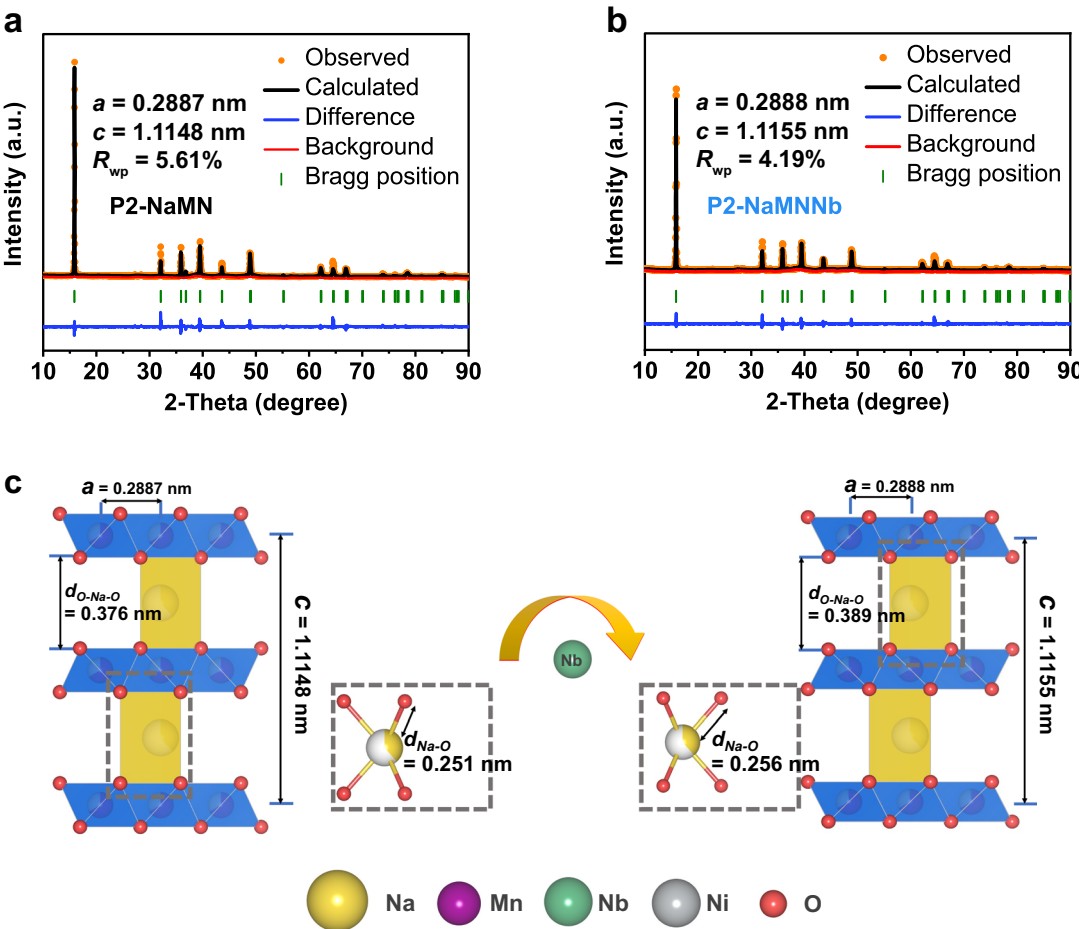

**Fig. 1 Structural characterization of the P2-NaMNNb and P2-NaMN cathode material. a, b** Rietveld refinements of X-ray diffraction measurements of P2-NaMN and P2-NaMNNb compounds. **c** Schematic diagram of the effect of Nb doping on crystal structure.

preconstructed layer is expected to be beneficial to improving structural stability upon Na$^+$ de/intercalation during the electrochemical tests[15,33–35]. As the enlarged views show in Fig. 2e, f (corresponding to the red and blue rectangular areas in Fig. 2d), rock-salt-like and spinel-like structures could be observed at the surface and the sub-surface of P2-NaMNNb, which may be attributed to the disordered cation arrangement[36]. Furthermore, the evolution of the crystal structure was probed with the assistance of line-scan intensity profiles (Fig. 2g, h). As depicted by the simulated lattice structure of the preconstructed layer (Supplementary Fig. 5), the distance between adjacent Mn (Ni/Nb)-I sites of the sub-surface is 0.85 nm, which is reduced to 0.822 nm at the surface due to the electrostatic attraction caused by the transition metals occupying the empty 16$c$ octahedral sites[37–40]. Moreover, direct chemical identification (magnified area in Fig. 2i) by using energy dispersive spectrometer (EDS) mappings and elemental line scan analyses further confirmed the elemental distribution from bulk to surface, from which we found that the concentration of Na in the preconstructed layer was relatively low but the concentration of Nb in the same area was relatively high compared to that at bulk area (Fig. 2i, j, Supplementary Fig. 6). It was proposed that the chemical compositions of the preconstructed layer for the Nb-doped sample are Mn, Ni, Nb, and O. Density functional theory calculations reveal the formation energies of Nb substitution for different atoms on both the bulk and surface (detailed information shown in Supplementary Fig. 7, Supplementary Table 4 and Supplementary Note 1), which indicates that Nb is prone to replace the Ni site in the bulk phase (−4.67 eV)

and the Na site in the surface phase (−5.59 eV). Therefore, in the synthesis process, the surface Na atoms could be readily replaced by Nb and cause a phase transition, while the rest of the Nb would be doped into the Ni site in the bulk phase.

The above microstructure analyses show that certain contents of Nb doping into the lattice of P2-NaMN could not only expand the d-spacing of the (00$l$) planes which is beneficial to Na$^+$ de/intercalation during cycling, but also form a Nb-rich preconstructed layer (3–5 nm) which might enhance the structural stability of the cathode material. It is known that electron-beam irradiation can provide a feasible stimulus that mimics the effect of electrochemical cycling[41]. Indeed, several reports have adopted electron-beam irradiation to estimate the structural stability of battery materials[42,43]. Here, in situ electron beam irradiation (300 keV) was performed to further investigate the mechanism of Nb doping on the structural stability enhancement of P2-NaMNNb. Interestingly, we found that P2-NaMNNb and P2-NaMN compounds exhibit different tolerance to electron beam irradiation (Supplementary Figs. 8 and 9). Under in situ electron beam irradiation, surface reconstruction and amorphization as well as material loss, can be observed for the P2-NaMN. In contrast, the P2-NaMNNb exhibits tolerability to electron beam irradiation due to the presence of the surface structures, which implies that such Nb-rich surface structures might be beneficial to the electrochemical stability of P2-NaMNNb.

**Electrochemical characterization of the P2-NaMNNb-based electrodes.** The electrochemical performance of P2-NaMNNb and

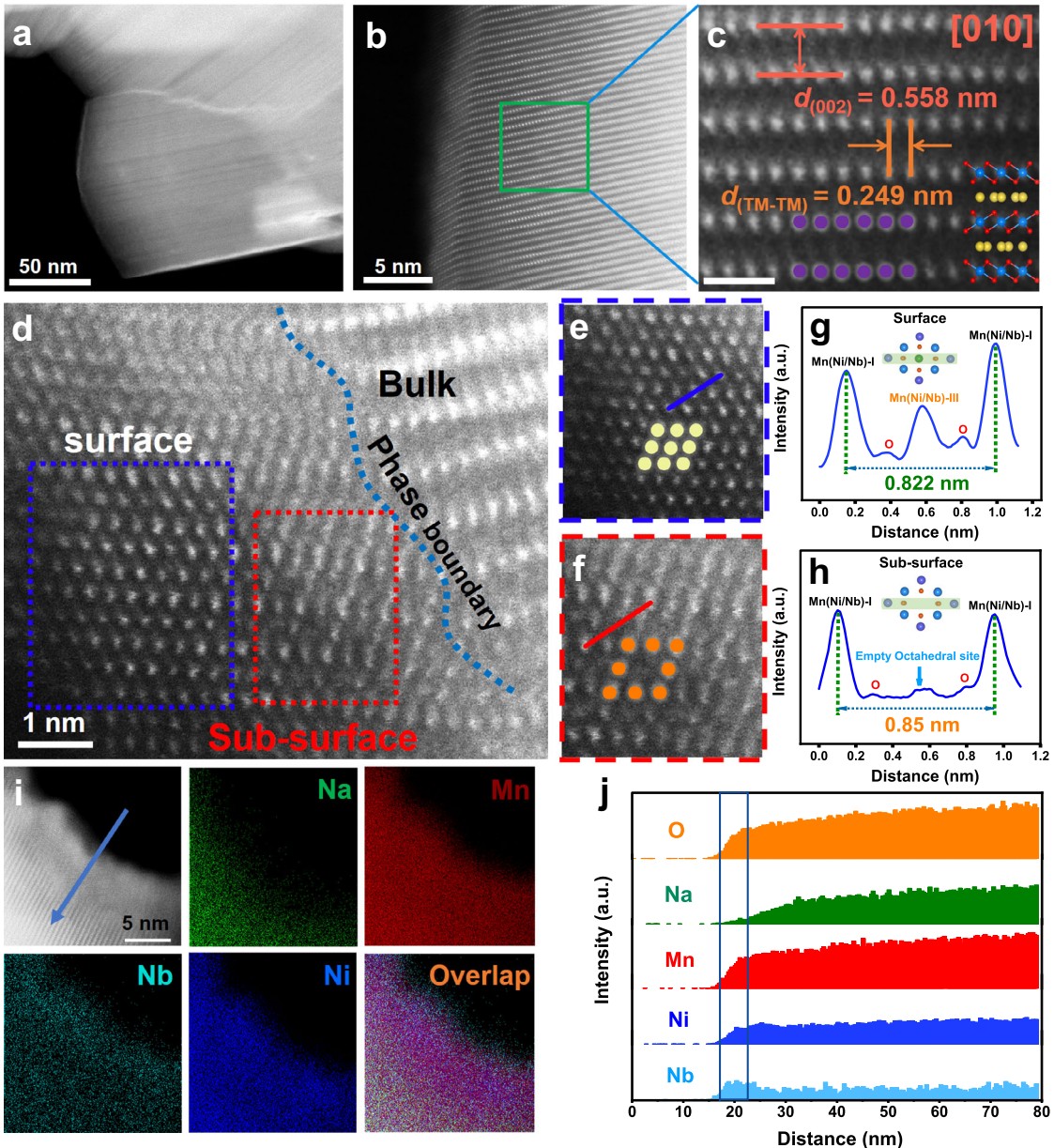

**Fig. 2 Transmission electron microscopy characterization of the P2-NaMNNb cathode material. a–c** STEM-HAADF images of P2-NaMNNb projected along the [010] zone axis. **d** A representative STEM-HAADF image of the interface between preconstructed layer and bulk phase of P2-NaMNNb. **e, f** Magnified views of the surface and sub-surface corresponding to the blue and red boxes in Fig. 2d. **g, h** Line profiles corresponding to the blue line in Fig. 2e and the red line in Fig. 2f. **i** STEM-EDS elemental maps of P2-NaMNNb compounds and corresponding chemical compositions with O (orange), Na (green), Mn (red), Ni (blue) and Nb (cyan-blue). **j** Energy Spectrum elemental line profiles of from surface to bulk phase corresponding to the blue line in Fig. 2i and corresponding chemical compositions with O (orange), Na (green), Mn (red), Ni (blue) and Nb (cyan-blue).

P2-NaMN were consistently tested at room temperature (RT, 25 °C) and LT (−40 °C). Cyclic voltammetry at RT for P2-NaMNNb and P2-NaMN in a two-electrode coin cell configuration (using Na metal as the counter/reference electrode) was carried out at a scan rate of 0.2 mV s$^{-1}$ (Fig. 3a). For P2-NaMNNb, three pairs of redox peaks were found at 3.12/3.2, 3.28/3.34, and 3.65/3.70 V vs. Na/Na$^+$, corresponding to the redox couples of Ni$^{2+}$/Ni$^{3+}$ and Ni$^{3+}$/Ni$^{4+}$, respectively[6,12,13,32,44]. Nb doping can efficiently reduce the polarization voltage (ΔV) between adjacent anodic and cathodic peaks. Indeed, ΔV is reduced from 97 to 61 mV, which agrees well with the charge/discharge profiles of P2-NaMNNb and P2-NaMN (Supplementary Figs. 10 and 11). The valance states of different elements in the selected potential range were verified through ex situ X-ray absorption spectroscopy (XAS) and ex situ X-ray photoelectron

spectroscopy (XPS) characterizations (Supplementary Figs. 12–14 and Supplementary Note 2–4). The electrochemical performance at −40 °C of the as-prepared samples was tested to verify the application feasibility at LT. Unlike the voltage polarization (264.7 mV) on CV curves for P2-NaMN, P2-NaMNNb shows a smaller voltage polarization (126 mV) at LT (Fig. 3b). The galvanostatic intermittent titration technique (GITT) (Fig. 3c, d) was employed to assess the Na$^+$ transport kinetics of P2-NaMNNb and P2-NaMN at both RT and LT, and the Na$^+$ diffusion coefficient ($D_{Na^+}$) was calculated according to the Eq. (2)[14,45] in "Electrochemical characterizations" section. It is promising that the as calculated $D_{Na^+}$ value of P2-NaMNNb doesn't fluctuate much from 25 °C (10$^{-11.36}$ ~ 10$^{-9.09}$ cm$^2$ s$^{-1}$) to −40 °C (10$^{-9.81}$ ~ 10$^{-9.29}$ cm$^2$ s$^{-1}$), confirming the excellent LT kinetics

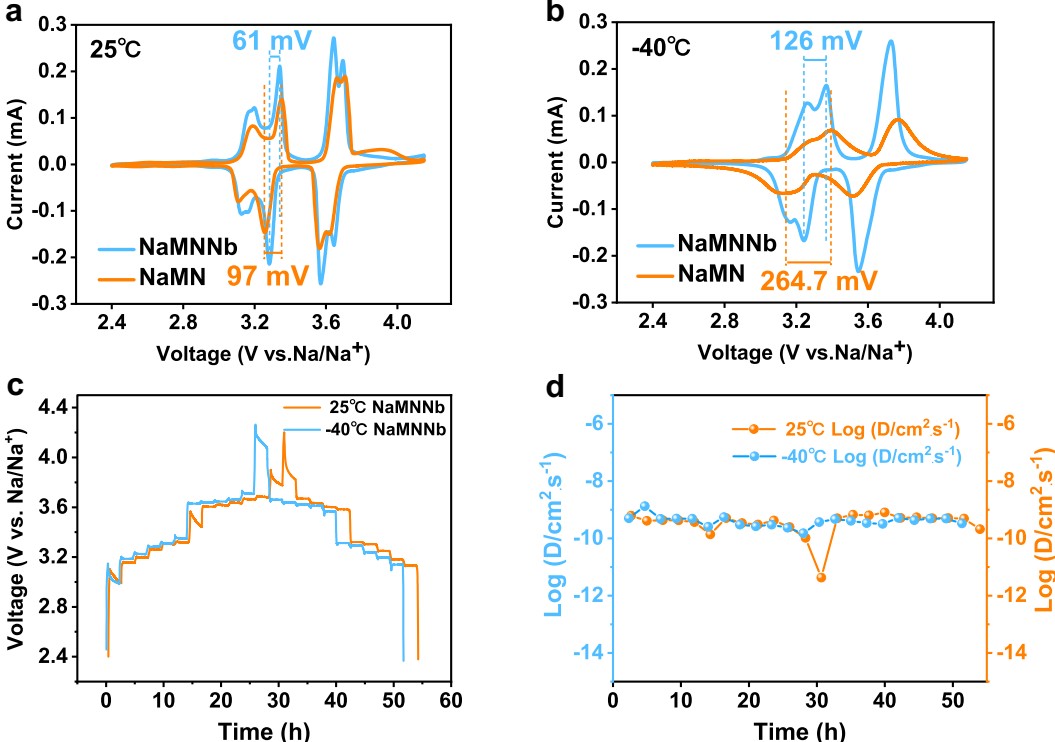

**Fig. 3 Electrochemical characterizations of the P2-NaMNNb and P2-NaMN cathode active material at various temperatures. a** CV curves of P2-NaMNNb and P2-NaMN at a scan rate of 0.2 mV s$^{-1}$ at 25 °C. **b** CV curves of P2-NaMNNb and P2-NaMN at a scan rate of 0.2 mV s$^{-1}$ at −40 °C. **c** GITT curves of P2-NaMNNb at 25 °C and −40 °C. **d** The corresponding sodium ion diffusion coefficient ($D_{Na}^+$) of P2-NaMNNb calculated from GITT formula at 25 °C and −40 °C.

of the material[12,13,22,46]. In contrast, the diffusion coefficients of P2-NaMN are unstable (Supplementary Fig. 15). Indeed, the P2-NaMN $D_{Na}^+$ range at 25 °C is between $10^{-11.64}$ and $10^{-10.09}$ cm$^2$ s$^{-1}$ while at −40 °C, the range between $10^{-11.43}$ and $10^{-10.06}$ cm$^2$ s$^{-1}$. In addition, a negligible average IR drop (4.25 mV) and voltage polarization (27.2 mV) are achieved for P2-NaMNNb at 25 °C, which are 10.5 and 16.1 mV at −40 °C respectively (Supplementary Fig. 16 and Supplementary Note 5), indicating good Na$^+$ transport kinetics even at LT.

Galvanostatic charge/discharge curves (GCD) of P2-NaMNNb from 92 mA g$^{-1}$ to 9.2 A g$^{-1}$ were tested at RT to investigate the electrochemical energy storage behavior of the Nb-doped cathode material (Fig. 4a, b). The as prepared P2-NaMNNb shows a specific capacity of 96.6 mAh g$^{-1}$ in the voltage range of 2.4–4.15 V at 92 mA g$^{-1}$ and 65.8 mAh g$^{-1}$ at 9.2 A g$^{-1}$ (in the same voltage range). The average Na‖P2-NaMNNb coin cell discharge voltage is approximately 3.4 V at 25 °C (Fig. 4b). In contrast, P2-NaMN shows a poorer rate capability, with only 81% capacity retention (i.e., 68.4 mAh g$^{-1}$ at 1.84 A g$^{-1}$) of that at 92 mA g$^{-1}$. The P2-NaMNNb shows a 98% capacity retention (i.e., 94.5 mAh g$^{-1}$) at 92 mA g$^{-1}$ and −40 °C considering cycling at 25 °C for the same specific current. At −40 °C, the Na‖P2-NaMNNb coin cell delivers a specific capacity of 83.6 mAh g$^{-1}$ and 62.9 mAh g$^{-1}$ at 920 mA g$^{-1}$ and 1.84 A g$^{-1}$, respectively (Fig. 4c, d). In contrast, P2-NaMN exhibits a discharge capacity of only 69.9 mAh g$^{-1}$ at 92 mA g$^{-1}$ (82% of that at RT) and delivers 59.7 and 39.2 mAh g$^{-1}$ at 184 mA g$^{-1}$ and 368 mA g$^{-1}$, respectively. It should be pointed out that the Nb doping enables the relatively high Ni content (0.31), thus maintains the voltage plateaus of Ni$^{2+}$/Ni$^{3+}$/Ni$^{4+}$ in the 4.15–3.3 V voltage range. Therefore, the complete solid-solution reaction is not always the necessities in perusing high rate performance for P2-type layered cathodes[13,14,28,45].

P2-NaMNNb exhibits a specific capacity of 182.1 mAh g$^{-1}$ at voltages of 1.5–4.15 V, which remains 88.8 mAh g$^{-1}$ at 5.52 A g$^{-1}$. However, to avoid the Jahn–Teller effect induced by Mn$^{3+}$/Mn$^{4+}$ at 1.5–2.2 V, suitable potential window of 2.4–4.15 V was selected to maximize the rate and cycling performance (Supplementary Fig. 17). We further compared the performance of P2-NaMNNb with P2-type cathode materials reported in the literature tested in different voltage ranges in terms of averaged cell discharge voltage, specific capacity, and rate performance (Supplementary Fig. 18 and Supplementary Table 5). It is found that the rate performance of P2-NaMNNb is well-positioned considering the similar materials from the literature both in large and narrow voltage ranges[12,13,46–49].

Subsequently, the long-term stability of P2-NaMNNb was tested at 920 mA g$^{-1}$ at 25 °C, with a capacity retention of 80% after 500 cycles (Supplementary Fig. 19). Under cycling conditions of −40 °C and 368 mA g$^{-1}$, the Na‖P2-NaMNNb coin cell retains a reversible discharge capacity of 69 mAh g$^{-1}$ after 1800 cycles (calculated capacity decay rate of 0.013% per cycle) (Fig. 4e–g, Supplementary Fig. 20). Conversely, Na‖P2-NaMN shows severe capacity loss at −40 °C and 920 mA g$^{-1}$ (Fig. 4c) and a discharge capacity retention of 77.8% after 215 cycles at RT (Supplementary Fig. 19). Furthermore, the complex impedance plot (see Fig. 4f) obtained via electrochemical impedance spectroscopy (EIS) measurements allows us to quantitatively identify the ion and charge transfer resistances of the electrodes[50]. An equivalent circuit model (inset of Fig. 4f) is applied to obtain the numerical values of the high-frequency Ohmic resistance ($R_0$), the solid electrolyte interphase (SEI) resistance ($R_1$) and the charge transfer resistance ($R_2$)[51]. The slight $R_1$ variation for the Na‖P2-NaMNNb coin cell after 1800 cycles at −40 °C demonstrates the retention of the favorable ionic conductive properties also at LT (Supplementary Fig. 21 and

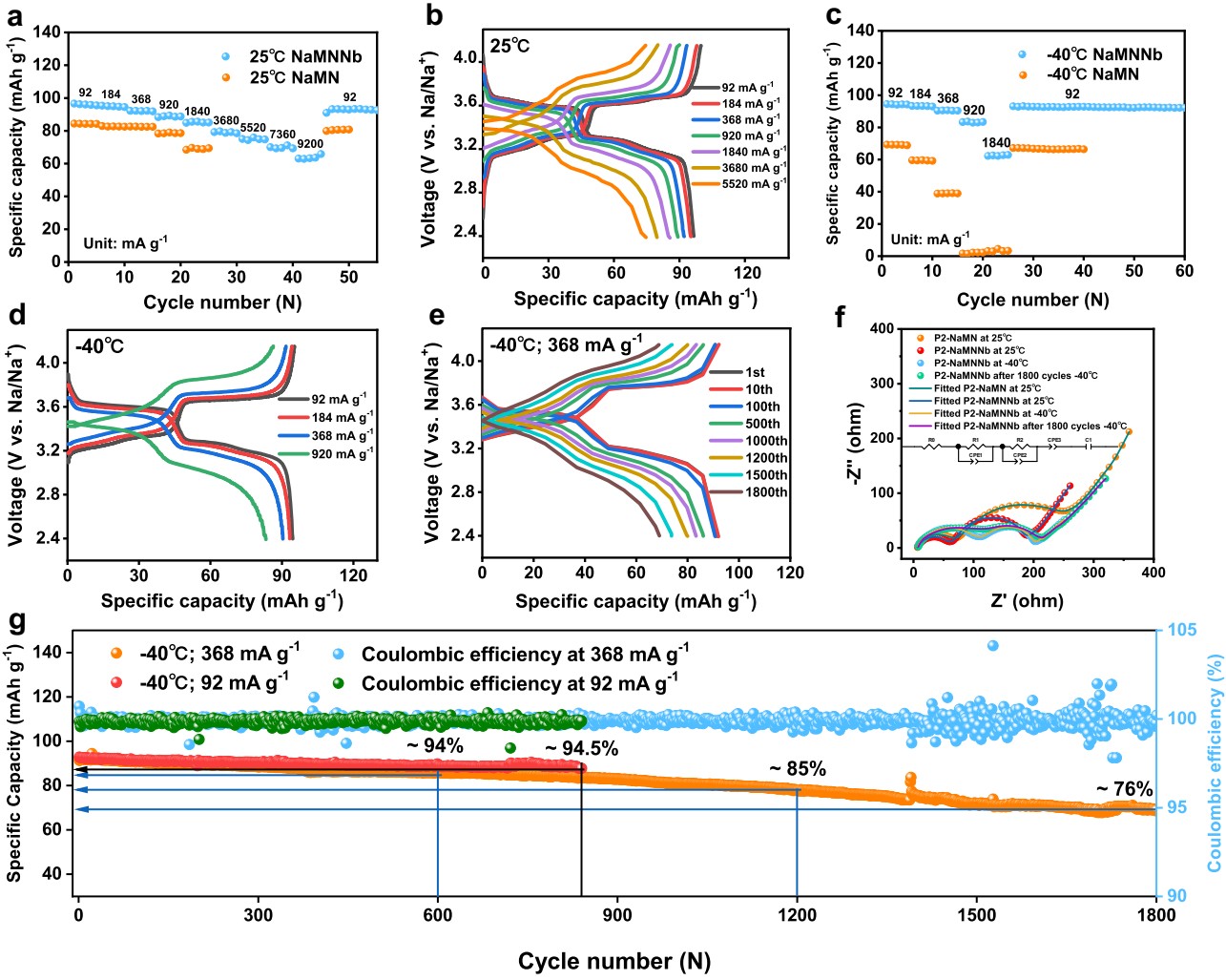

**Fig. 4 Electrochemical energy storage performance of Na||P2-NaMNNb and Na||P2-NaMN coin cells at various temperatures. a** The rate performance comparison of Na||P2-NaMNNb and Na||P2-NaMN at 25 °C. **b** Charge and discharge curves of Na||P2-NaMNNb at various rates at 25 °C. **c** The rate performance comparison of Na||P2-NaMNNb and Na||P2-NaMN at −40 °C. **d** Charge and discharge curves of Na||P2-NaMNNb at various rates at −40 °C. **e** Charge and discharge curves of Na||P2-NaMNNb coin cells at various cycle numbers at −40 °C and 368 mA g⁻¹. **f** EIS curves of P2-NaMNNb and P2-NaMN at 25 °C and −40 °C. **g** Long-term cycling stability at rates of 92 and 368 mA g⁻¹ at −40 °C.

Supplementary Table 6). To investigate the LT behavior of NaPF$_6$ in diglyme, we placed the electrolyte in an environment of −40 °C for 72 h. As shown in Supplementary Fig. 22, the electrolyte did not show solidification even after 72 h at −40 °C. Linear scan voltammetry (LSV) measurements were carried out at 0.2 mV s⁻¹ using a Na||Al asymmetric cell (Supplementary Fig. 23). The results demonstrate that electrolyte did not show oxidation until 4.65 V vs. Na⁺/Na at −40 °C. It is thus possible to assume that the Nb doping and the resulted surface preconstructed layer of this P2-type cathode with high Na-content improves the electrochemical energy storage behavior of SIBs at 25 °C and −40 °C.

**Ex situ and in situ structural characterization of the P2-NaMNNb-based electrodes.** In situ and ex situ measurements were carried out on P2-NaMNNb-based electrodes (in coin cell configuration using Na metal as counter/reference electrode) during or after cycling in the 2.4–4.15 V cell voltage range. It is worth noting that, as Na⁺ ions are extracted during the charging process, the (002) and (004) diffraction peaks sequentially shift to lower angles during the charge process due to the increased electrostatic repulsion between the oxygen anion of Na layers[14,52].

Still in the charging process, the (100), (102), and (103) peaks move to higher angles, which may be attributed to the reduction of in the ionic radius caused by electrochemical oxidation of transition metals. After the full cycle, all the characteristic diffraction peaks revert to their original initial position without any new phase observed (Fig. 5a, Supplementary Fig. 24), with low c-axis variation (~2%), and volume change (~1.7%). In contrast, a peak is observed at 16.4° for P2-NaMN, corresponding to the P2' phase (*cmcm*, JCPDF no. 27-752). This phase is a distorted hexagonal P2 phase with negligible variation of β from 90° to 90.68° and often appeared under 2.7 V, which could cause poor rate capability and severe capacity decay upon cycling[11,23,53] (Fig. 5b, Supplementary Fig. 25).

The water contained in the non-aqueous electrolyte solution (in the order of ppm as reported in the "Methods" section) could react with air-instable Na$_x$TMO$_2$ by inserting into the Na layer or exchanging Na⁺ with H⁺, which would form an unfavorable hydration phase upon cycling, leading to a suppressed Na⁺ (de) intercalation ability and insufficient electrochemical performance[54–56]. The P2-NaMNNb demonstrates a small hydration peak at 11.6° after cycling, which is much stronger for P2-NaMN (Fig. 5b). For such a reason, the microstructure

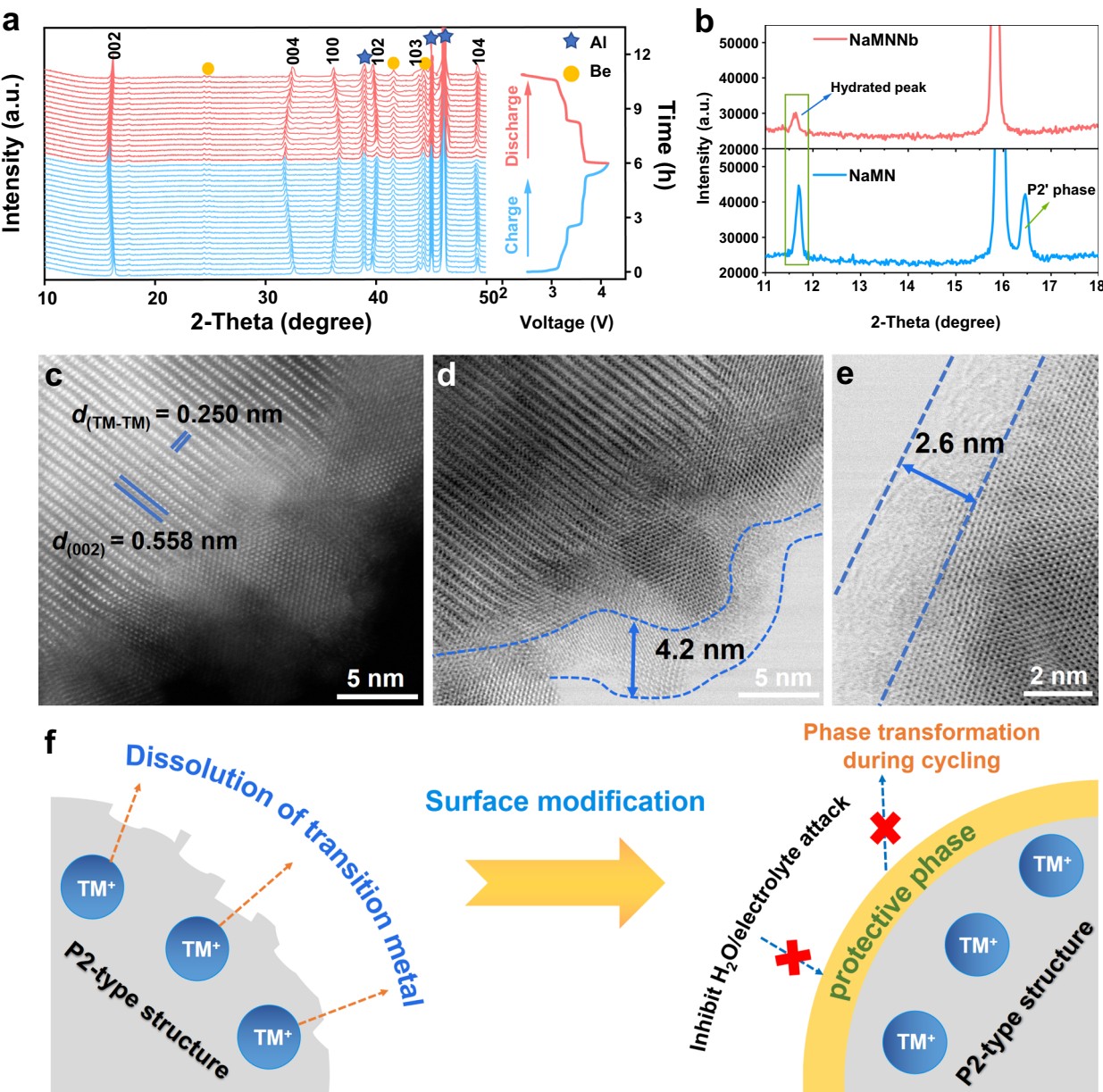

**Fig. 5 In situ and ex situ structural characterizations of the P2-NaMNNb-based electrodes. a** In situ XRD patterns using Swagelok cell corresponding to the charge and discharge curves between 2.4–4.15 V at 20 mA g$^{-1}$ and 25 °C. **b** Comparison of (002) diffraction patterns of ex situ XRD patterns of P2-NaMNNb and P2-NaMN after 30 cycles at 92 mA g$^{-1}$. **c** The AC-HAADF-STEM image and **d**, **e** annular bright field (ABF-STEM) images of the P2-NaMNNb after 30 cycles at 92 mA g$^{-1}$ and 25 °C in coin cell. **f** Schematic of the protective effect on the bulk structure.

evolution after several cycles was also investigated via ex situ TEM measurements. The HAADF-STEM reveals that the thickness of the surface layer of the P2-NaMNNb remains 3–5 nm after 30 cycles, and the bulk structure remains almost unchanged with a $d$-spacing of 0.558 nm for the TM-layer and 0.250 nm for the interlayer spacing of TM-atoms (Fig. 5c). Moreover, a robust CEI of approximately 2~5 nm is formed on the surface of the P2-NaMNNb cathode after 30 cycles. (Fig. 5d, e and Supplementary Fig. 26a), and such a thin and stable CEI layer could not only reduces the charge transfer resistance, but also improves the rate capability and LT performance. In contrast, P2-NaMN encounters a severe surface degradation, and the thickness of CEI is measured as 18.2 nm (Supplementary Fig. 26b). In addition, the EELS spectra show that the intensity of the pre-edge

of O K-edge for the P2-NaMNNb sample after 30 cycles is consistent with that of the pristine sample (Supplementary Figs. 27 and 28 and Supplementary Note 6). The variation in the L-edge of Mn and Ni from the bulk to the surface is also in alignment with that in the pristine one. The surface composition of on the Na anode was also examined with energy dispersive X-ray spectroscopy (EDX), whereby the P2-NaMNNb shows negligible TM signals of 0.06 at% and 0.04 at% for Mn and Ni, which are 0.3 at% and 0.25 at% for P2-NaMN respectively. This indicates that the preconstructed surface layers can effectively prevent the bulk transition metals from dissolving (Supplementary Fig. 29). Therefore, the preconstructed layer plays a critical role in inhibiting the P2-P2' phase transition and surface degradation, forming a stable and thin CEI layer as well as

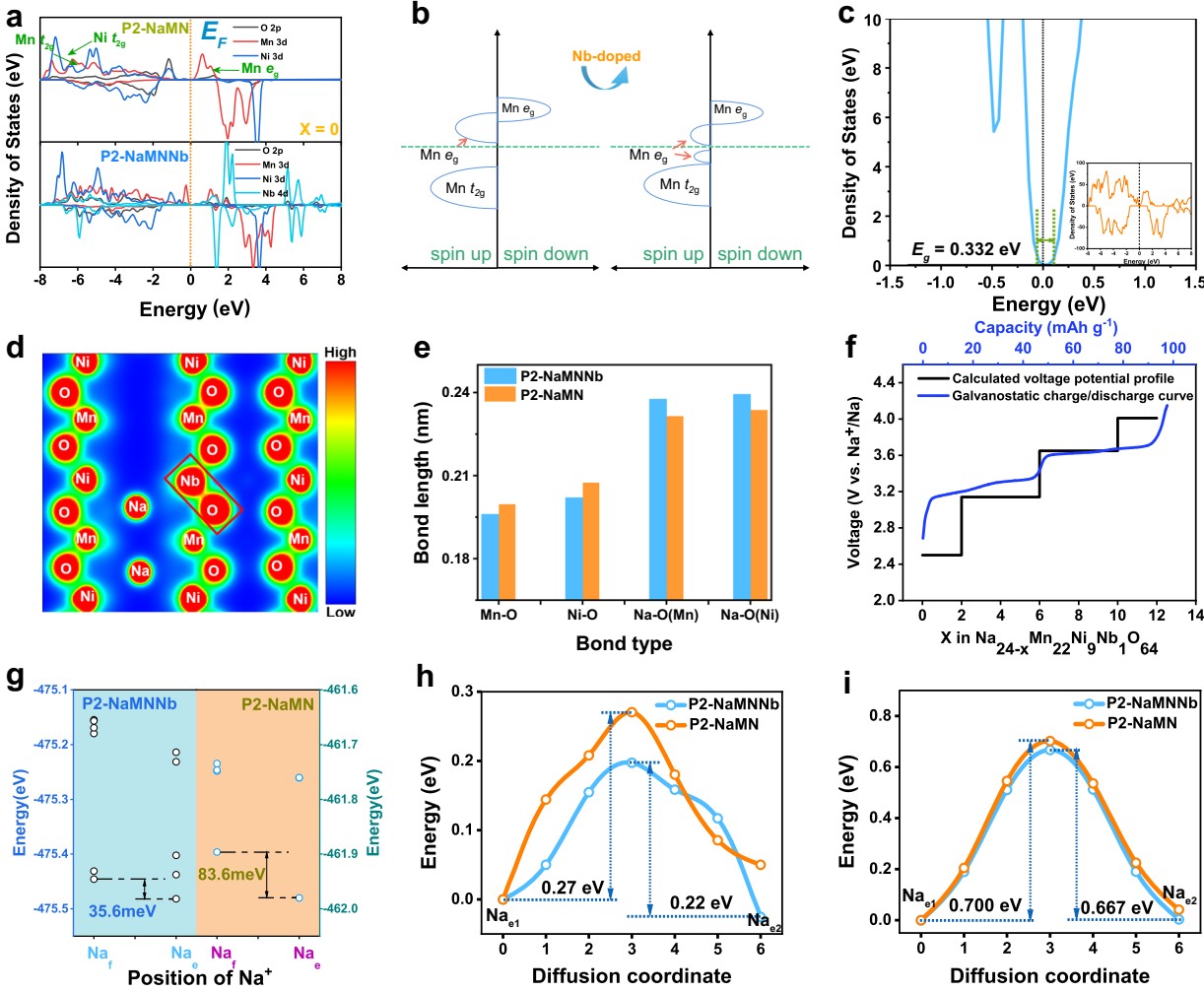

**Fig. 6 Theoretical investigations on the evolution of the P2-NaMNNb and P2-NaMN electronic structures. a** Partial density of states (pDOS) of O 2*p*, Mn 3*d*, and Ni 3*d* and Ni 4*d* orbitals with x = 0. **b** Schematic pDOS of P2-NaMN and P2-NaMNNb with x = 0. **c** The total density of state of $Na_{24}Ni_9Mn_{22}Nb_1O_{64}$. **d** Charge density distribution for P2-NaMNNb (visualized by Visualization for Electronic and Structure Analysis (VESTA)), the red regions stand for high charge density and blue regions for low charge density. **e** The comparison of average bond length for TM-O and Na-O of P2-NaMN and P2-NaMNNb. **f** Comparison of calculated voltage potential and galvanostatic charge/discharge curves at 92 mA g$^{-1}$ and 25 °C in coin cell. **g** Calculated energy difference between the $Na_e$ and $Na_f$ site for P2-NaMNNb (blue background) and P2-NaMN (orange background). Calculated Na$^+$ ion diffusion pathways of P2-NaMNNb with **h** Na deficient status, **i** Na rich status, and the corresponding Na$^+$ migration energy barriers of P2-NaMN and P2-NaMNNb.

preventing water molecules from entering the crystal lattice (Fig. 5f), which significantly promotes the rate performance and cycling stability of P2-NaMNNb.

**Theoretical investigation of the Na-ion storage mechanism in P2-NaMNNb.** Density functional theory (DFT) calculations were conducted to investigate the (de)intercalation chemistry of Na$^+$ in P2-NaMNNb. A reasonable $4 \times 4 \times 1$ ($Na_{24-x}Ni_{10}Mn_{22}O_{64}$) supercell with/without Nb substitution was constructed. After the structural optimization, the electronic structure, charge density, and the Na$^+$ migration barrier were calculated. Figure 6a shows the partial density of states (pDOS) for transition metal 3*d* orbitals and oxygen 2*p* orbitals. It is noticed that after Nb doping, the Mn $e_g$ orbital splits into two peaks and one of them moves below the Fermi level to maintain charge conservation and reduce the band gap ($E_g$) from 0.500 eV to 0.332 eV according to the total density of states, underlining an enhanced electronic conductivity (Fig. 6b, c, Supplementary Fig. 30)[57,58]. The charge density and average bond lengths of TM-O and Na-O reflect that the interaction between TM and O is more intense (Fig. 6d, e) after Nb

doping[59–61]. The introduction of relatively short Nb-O bond results in increasing the bond length of Na-O but shortening that of TM-O, thus enhancing the bond energy of TM-O to maintain structural stability, which is in good accordance with the XRD refinement. The redox voltage potentials of P2-NaMNNb, corresponding to different amounts of Na, are also simulated and the results are compared with experimental galvanostatic charge/discharge curves at 92 mA g$^{-1}$ and 25 °C in coin cells. (Fig. 6f). It is seen that the calculated platform (black line) is in good accordance with the experimental blue line. Furthermore, the calculated total DFT energies of the two kinds of Na sites in the P2-type structure illustrate that energy difference between the $Na_e$ and $Na_f$ sites can be reduced in P2-NaMNNb (from 83.6 meV to 35.6 meV), implying that the hoping is easier when Nb is involved (Fig. 6g)[45].

To better understand the rate capability and LT electrochemical energy storage behavior of P2-NaMNNb, the nudged elastic band (NEB) method was performed to evaluate the energy barrier ($E_a$) for Na$^+$ migration. Considering that the energy of $Na_e$ site is lower than $Na_f$ site in Na$_x$TMO$_2$, we designed a diffusion path from $Na_{e1}$ to $Na_f$, and then $Na_f$ back to $Na_{e2}$ (Fig. 6h). P2-

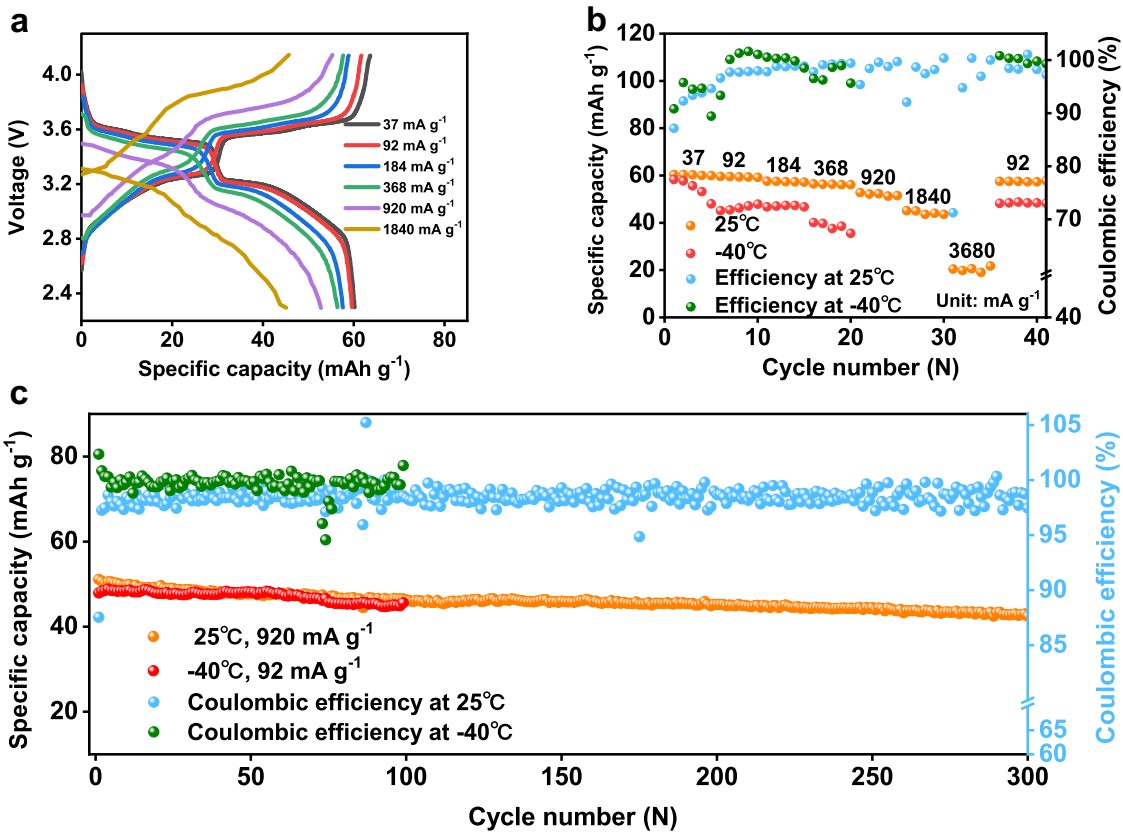

**Fig. 7 Electrochemical energy storage performance of the pre-sodiated hard carbon||P2-NaMNNb coin cells at various temperatures. a** The electrochemical curves of the full-cell with various specific current. **b** The rate performance of the full-cell at 25 °C and −40 °C. **c** Cycling performance in a rate of 920 mA g⁻¹ at 25 °C and 92 mA g⁻¹ at −40 °C.

NaMNNb encounters a small $E_a$ of 0.22 eV for Na⁺ diffusion compared to 0.27 eV for P2-NaMN. Besides, the diffusion coefficient of adjacent Na⁺ sites can be reckoned by the following equation:

$$D = a^2 v^* \exp\left(\frac{-Ea}{k_B T}\right) \tag{1}$$

where $a$, $v^*$, $Ea$, $k_B$, and $T$ stand for the hopping distance between two adjacent sites, the hopping frequency ($10^{13}$ s⁻¹), the calculated largest barrier via each path, Boltzmann's constant, and the temperature, respectively[62]. The calculated $D_{Na}^+$ of P2-NaMNNb is $1.08 \times 10^{-6}$ and $9.74 \times 10^{-8}$ cm² s⁻¹ at RT and LT, respectively, which is one order of magnitude higher than that of P2-NaMN ($1.60 \times 10^{-7}$ and $8.59 \times 10^{-9}$ cm² s⁻¹). Therefore, the Nb doping can efficiently boost the reaction kinetics at both RT and LT. In addition, the $Ea$ for Na⁺ migration at Na-rich status (with the generated model of one Na vacancy) is also calculated, which is 0.667 eV for P2-NaMNNb, and 0.7 eV for P2-NaMN (Fig. 6i). It is worth noting that, as compared to the Na-rich status, the calculated $D_{Na}^+$ is much smaller (0.667 vs. 0.22 eV) at Na-deficient state, corresponding to the high charge/discharge potential in the GLD curves. Therefore, controlling the potential window within a relatively high voltage range (2.4–4.15 V) would benefit the rate performance of the electrode material while avoiding the Jahn–Teller effect of Mn³⁺.

**Electrochemical characterizations of the Hard Carbon||P2-NaMNNb coin cells**. We also tested the P2-NaMNNb-based positive electrode in combination with a pre-sodiated hard carbon-based negative electrode in coin cell configuration using the 1 M NaPF₆ in diglyme electrolyte solution (Fig. 7a and Supplementary

Fig. 31). It should be pointed out that the Na-ion storage behavior of the hard carbon-based electrode was preliminarily investigated in Na||hard carbon coin cell at 25 °C and 50 mA g⁻¹ where a reversible sodiation capacity of approximately 290 mAh g⁻¹ at the 2nd cycle was obtained (Supplementary Fig. 31b, c). The full-cell delivered a specific capacity of 60.4 mAh g⁻¹ in the voltage range of 2.3–4.14 V at 37 mA g⁻¹ and 25 °C, corresponding to a specific energy of 202 Wh kg⁻¹. It should be noted that, both the specific capacity and the specific energy of the hard carbon||P2-NaMNNb coin cells are calculated based on the total loading mass of the active materials on both electrodes. Therefore, the specific capacity here is lower than that of the Na||P2-NaMNNb coin cells even at the same testing rates. The hard carbon||P2-NaMNNb coin cell can be efficiently cycled up to 3.68 A g⁻¹ at 25 °C delivering a discharge capacity of approximately 20 mAh g⁻¹ demonstrating a specific power (based on the active material masses of both negative and positive electrodes) of 7.75 kW kg⁻¹ (Fig. 7b, Supplementary Fig. 32), which is calculated based on the electrochemical performance of the full-cell. Furthermore, the hard carbon||P2-NaMNNb coin cell shows a discharge capacity retention of about 84% after 300 cycles at 920 mA g⁻¹ and 25 °C (Fig. 7c). The hard carbon||P2-NaMNNb coin cell tested at −40 °C and 37 mA g⁻¹ delivers a specific capacity of about 58 mAh g⁻¹ (Fig. 7b) and specific energy of around 188 Wh kg⁻¹ (Supplementary Fig. 32) which translates into capacity retention of 95.5% compared to the same cycling conditions at 25 °C. Moreover, as can be seen in Fig. 7c, the hard carbon||P2-NaMNNb coin cell tested at −40 °C and 92 mA g⁻¹ demonstrates capacity retention of approximately 89% after 100 cycles. These battery performances are well-positioned compared to the state-of-the-art literature of Na-ion cells[49,63–65], as shown in Supplementary Fig. 32, Supplementary Tables 7 and 8.

## Discussion

The good low-temperature performance and rate capability of the P2-NaMNNb-based electrodes could be explained taking into account various aspects as mentioned below: (I) DFT calculations confirm that the Nb doping could reduce the electronic band gap and improve the electrical conductivity of the material, and meanwhile the $Na^+$ diffusion mobility is promoted with suppressed energy barrier especially at the Na-deficient status. Therefore both the electronic and ionic mobility can be significantly enhanced. (II) The doped Nb induces the formation of an atomically-thin interlayer, which could prevent $H_2O$ and solvent molecules from entering the crystal lattice, and suppress the structural distortion and P2-P2' phase transition, inhibit the loss of transition metals, and form a robust CEI layer to achieve stable $Na^+$ (de)intercalation. (III) The Nb (0.02) substitution enables a relatively high Ni content (0.31), thus maintaining the voltage plateaus of $Ni^{2+}/Ni^{3+}/Ni^{4+}$ at 4.15~3.3 V. Therefore, the complete solid-solution reaction is not always the necessities to achieve high rate performance for P2-type layered cathodes. (IV) The relatively high Na content (0.78) enables the crystal structure to remain tight at the Na deficient status, which also benefits the fast $Na^+$ (de)intercalation at the Na-deficient status.

In summary, we have successfully constructed a high sodium content P2-$Na_{0.78}Ni_{0.31}Mn_{0.67}Nb_{0.02}O_2$ material with a modulated bulk and surface structure. The Nb substitution for Ni could regulate the bond length of TM-O and Na-O, and facilitate the high $Na^+$ mobility and low activation energy barriers especially at the Na deficient status. Moreover, the Nb doping induced surface preconstruction could form a stable CEI film and prevent phase transformation and surface degradation, thus stabilizing the $Na^+$ (de)intercalation reaction during the cycling process. The P2-$Na_{0.78}Ni_{0.31}Mn_{0.67}Nb_{0.02}O_2$-based electrode delivers a reversible discharge capacity of about 66 mAh $g^{-1}$ and an average discharge voltage of 3.4 V when tested in combination with a Na metal electrode at 25 °C and 9.2 A $g^{-1}$ in the 2.4–4.15 voltage range. The cathode material can also deliver good battery performances at −40 °C in combination with a pre-sodiated hard carbon-based negative electrode (e.g., a specific capacity of about 58 mAh $g^{-1}$ at 37 mA $g^{-1}$).

## Methods

**Materials**. $Na_{0.78}Ni_{0.31}Mn_{0.67}Nb_{0.02}O_2$ and $Na_{0.78}Ni_{0.32}Mn_{0.68}O_2$ samples were prepared by co-precipitation and the solid phase sintering method. Specifically, 0.017 mol the chemicals of Nickel acetate (Aladdin, >99.0%) and 0.036 mol manganese acetate (Aladdin, >99.0%) were firstly added into distilled water with a stoichiometric ratio under magnetic stirring at 25 °C for 60 min to obtain a homogeneous solution, marked as A. Weighing 0.07 mol $Na_2CO_3$ and dissolve it under magnetic stirring in 50 ml deionized water, marked as B. Then, B was dropped into A. The solution was stirred for 12 h, then washed with 50 ml water or 40 ml ethanol for at least 3 times, and dried at 80 °C in a vacuum oven for 6 h to obtain the carbonate precursor. Next, mixed the carbonate precursor of 0.01 mol, with 0.004 mol $Na_2CO_3$ and $1.27 \times 10^{-4}$ mol $Nb_2O_5$ into planetary ball mills and milling with high energy in air with 1000 rpm at 25 °C for 6 h. The power was calcined at 500 °C for 10 h and then sintered in corundum crucible at 900 °C for 12 h in Muffle furnace in air atmosphere with heating rate of 5 °C / min and cooled down to room temperature. Polyvinylidene fluoride (PVDF, 99.5%, Alfa Aesar), N-Methyl pyrrolidone (NMP, 99.5%, Alfa Aesar), hard carbon (average particle size of 1.1 nm; in-house prepared via bottom-up ZnO-assisted bulk etching strategy[66]), acetylene black (99.5%, Alfa Aesar, average particle size: 300 nm). Al foil (20 μm, 99.5%, Alfa Aesar) and Cu foil (18 μm 99.5%, Alfa Aesar), Na metal foil (99%, Alfa Aesar), Na discs were rolled into thin pieces and punched out of the large Na metal chunks (diameter: 16 mm, thickness: 0.5 mm). The electrolyte was 1 mol $L^{-1}$ $NaPF_6$ dissolved in 100% diglyme ($H_2O \leq 20$ ppm). The electrochemical measurements at low temperature were carried out in chamber (manufactured by Guang Pin Bo Gong test equipment in Shanghai), which can regulate temperature from −60 to 80 °C, the error of temperature is not exceeding to ±1 °C (Supplementary Fig. 33).

**Physicochemical characterizations**. Inductively coupled plasma atomic emission spectrometry (ICP-AES, PERKINE 7300DV) was used to investigate the composition of elements in samples. The samples are dissolving in nitric acid and the composition of different elements were quantified by elemental characteristic spectra and their characteristic peak intensities. X-ray powder diffraction (PXRD)

data were performed by utilizing Rigaku smartlab X-ray diffractometer with Cu Kα radiation ($\lambda = 0.15418$ nm). In situ XRD studies, Be piece (thickness: 2 mm) and Al foil (thickness: 14 μm) were prepared as X-ray-transparent windows with a Swagelok cell at 20 mA $g^{-1}$ during charge and discharge process at 25 °C. Rietveld refinement of X-ray diffraction were analyzed by FullProf Suite. The morphology of all samples are investigated by a scanning electron microscope (SEM, JEOL JSM-6701F). A spherical aberration-corrected (Cs-corrected) scanning transmission electron microscopy (STEM) performed High angle annular dark-field (HAADF) analysis and operated at 300 kV (JEM Grand ARM300F, JEOL). X-ray absorption near-edge structure (XANES) measurements were performed at Canadian Light Source Inc, Canada, using IDEAS and SXRMB beamlines. Ge (220) (IDEAS) and Si(111) (SXRMB) double crystal monochromators were used to cover the energy range of Mn and Ni K-edge in transmission mode.

**Electrochemical characterizations**. The electrode slurry consisted of active material, acetylene black and polyvinylidene fluoride (PVDF) with the mass ratio of 8: 1: 1, dissolved in N-Methyl pyrrolidone (NMP) and stirring for 6 h, then coating on an Al foil with loading density of 2–3 mg $cm^{-2}$ and dried under vacuum at 80 °C for 12 h. The diameter of cathode disc is 14 mm The electrolyte was 1 mol $L^{-1}$ $NaPF_6$ dissolved in 100% diglyme, and the Glass fiber (GF/D, Whatman, diameter: 20 mm, average pore size: 2.7 μm) was used as the separator. For full-cell equipment, the hard carbon anode was blending with PVDF 10% and acetylene black 10%, then coating on copper foil with average loading density of 0.7–1.2 mg $cm^{-2}$ and dried under vacuum at 80 °C for 12 h. The ratio between negative and positive electrode material was set as around 1.2–1.4 (capacity ratio). After that, the discs were punched out and equipped half-cells with sodium metal, then pre-sodiated for 3 cycles within 0.01–2 V and stopped at 0.2 V in charge state to form the SEI film and inhibit irreversibility during sodiation process. All the 2032 coin cells were fabricated in an Ar-filled glovebox with $H_2O$ and $O_2 < 0.01$ ppm. For ex situ XAS and XPS measurements, the electrodes after disassembling process were used diglyme to wash for one time and dried in an argon-filled glove box, then the samples were transferred and sealed in a specific testing sample chamber in an argon-filled glove box for measurements. Cyclic voltammetry (CV) results were collected at 0.2 mV $s^{-1}$ from electrochemical workstation (CHI760E). Linear scan voltammetry (LSV) measurements were carried out at 0.2 mV $s^{-1}$ using a Na||Al asymmetric cell in the voltage range of 3.2–4.8 V. EIS measurements were carried out at the open circuit potential from 0.01 Hz to $10^5$ Hz at 25 °C with a potentiostatic signal. The number of data points was 12 per decade, and the amplitude of the signal was 5 mV. The assembled cells were aged for 12 h to ensure that the system was fully wetted. The calculation method of specific power and specific energy are presented in Supplementary Note 7. Land BT2000 battery system (Wuhan, China) are used to test the charge and discharge process. $D_{Na}^+$ from GITT results is calculated by the following equation

$$D_{Na}+ = \frac{4}{\pi\tau}\left(\frac{m_B V_M}{M_B S}\right)^2 \left(\frac{\Delta E_S}{\Delta E_\tau}\right)^2 \tag{2}$$

Where $D$ is the diffusion coefficient of $Na^+$ in cathode, $V_M$ ($cm^3$ $mol^{-1}$) is the molar volume, $m_B$ and $M_B$ are molecular weight and relative molar weight of cathode material. $S$ (1.13 $cm^2$) is the surface area of the electrode. $\tau$ is the time for the applied current during the galvanostatic intermittent titration. $\Delta E_S$ and $\Delta E_\tau$ are the state of steady voltage and total variation of the battery voltage E during the current pulse.

**Computational methods**. DFT calculations were performed with the Vienna ab-initio simulation package (VASP)[67]. The calculations employed the PBE exchange-correlation functional, which is a generalized gradient approximation (GGA) method. The GGA + $U$ method was used because GGA cannot correctly reproduce the localized electronic states of the transition metal oxide materials. The $U$ values for Mn, Ni, Nb were 3.9, 6.9, 1.5 eV. Plane-wave projector augmented wave (PAW) pseudo-potentials were used to represent the core electrons. A plane-wave energy cutoff was 520 eV using for all calculations. NEB calculation is performed to find out the energy barrier for $Na^+$ migration[68]. Two models were constructed with single Na and one vacancy for $Na^+$ diffusion to avoid electrostatic repulsion between different Na ions[69]. All atoms were allowed to relax within the fixed lattice parameters during NEB process. the calculations of bulk phase were performed on $4 \times 4 \times 1$ supercells of a P2 type $Na_{24-x}Ni_9Mn_{22}Nb_1O_{64}$ structure. A $4 \times 4 \times 1$ supercell of (−103) plane was conducted with the thickness of vacuum of 1.5 nm. The average redox potential (V) is calculated by the equation, where E stands for the configuration with lowest formation energy of $Na_xMn_{0.67}$-$Ni_{0.31}Nb_{0.02}O_2$ at each composition, F is the Faraday constant.

$$V = \frac{E\left[Na_{x2}\left[Mn_{0.67}Ni_{0.31}Nb_{0.02}\right]O_2\right] - E\left[Na_{x1}\left[Mn_{0.67}Ni_{0.31}Nb_{0.02}\right]O_2\right] - (x_1 - x_2)E(Na)}{(x_2 - x_1)F}$$

(3)

## Data availability

The data supporting the findings of this study are available from the authors upon reasonable request.

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

## Acknowledgements

We thank the financial support from the National Natural Science Foundation of China (22179077, 51774251), Shanghai Science and Technology Commission's "2020 Science and Technology Innovation Action Plan" (20511104003), Shanghai Natural Science Foundation (21ZR1424200), Hebei Natural Science Foundation for Distinguished Young Scholars (B2017203313), Talent Engineering Training Funds of Hebei Province (A201802001), the Canadian Light Source (CLS). CLS is supported by the NSERC, NRC, CIHR of Canada, the University of Saskatchewan.

## Author contributions

Y.-F.Z. proposed and supervised the project. Q.-H.S., R.-J.Q., and Y.-F.Z. designed the experiments. Q.-H.S., X.-C.F. carried out the experimental study. Q.-H.S., X.-C.F., and J.W. performed the DFT calculation. R.-J.Q. carried out the TEM study. X.W. and Y.L. performed the in situ XRD measurements. Z.-P.Y., Q.Q.L, X.-G.L., and J.J.Z and other authors all participated in the analysis of the experimental results.

## Competing interests

The authors declare no competing interests.
