## [Peer Review File · Nature Communications]

Niobium-doped layered cathode material for high-power and low-temperature sodium-ion batteries.Reviewers' Comments:

Reviewer #1:

Remarks to the Author:

The manuscript presents an idea of Nb-doped well-known Na-Mn-Ni-O system and will be helpful for researchers to select electrode materials for Sodium-ion batteries. The presented capacity of NaMNNb is relatively low, taking into account other layered oxides like Na-Mn-Fe-O or Na-Co-Fe-O systems. However, the material demonstrates high cycling stability. Below I listed my comments. I recommend a major revision.

1. The authors indicated that nickel participates in the redox reactions during cell operation. How about Mn and Nb? Are these elements electrochemically inactive?
2. How can you explain such low capacity, comparing with other well-known sodium layered oxides, e.g. $\text{Na}_{0.67}\text{Mn}_{0.5}\text{Fe}_{0.5}\text{O}_2$?
3. In Fig.3b there is lack of capacity value for C-rate 20C - 50C for NaMN. Is this because NaMN did not work in such high current range?
4. The authors showed only one charge/discharge curve for every current rate for the 25oC. How about the first cycle, especially the charging process? Does it possess abnormal character or is it similar to these shown in Fig. 3c?
5. Why did the authors mention the 1C value in the bracket? This value is multiplication of 96.6 mAh/g and does not mean the actual value obtained by the cell. Or maybe this value is theoretical value calculated for the electrode mass? An explanation is necessary.
6. The authors conducted the in-situ XRD measurement to have an insight into structural evolution of NaMNNb cathode material during cell operation. But, what was the base for identifying some peaks (e.g. 16.4 2theta) as P2' phase? P2-type structures are known that can transform to other types, like O2, OP4 etc. Did you evaluate it by Rietveld refinement? In the manuscript, as well as in Supplementary information, I did not find suitable explanation. You should also add to the text the JCPD number of the phases you used.
7. In the experimental part in "Electrochemical measurement": there is a lack of chemical substances manufacturers. Moreover, what kind of test cell cases did you use?
8. Please, check your manuscript, because some typos and grammar mistakes are present.

Reviewer #2:

Remarks to the Author:

Shi et al. present an interesting property of P2- $\text{Na}_{0.78}\text{Ni}_{0.31}\text{Mn}_{0.67}\text{Nb}_{0.02}\text{O}_2$ layer compound that was active even at -40 oC, using an electrolyte of 1M NaPF₆ in diglyme in Na half- and full cells. A similar compound, $\text{Na}_{2/3}\text{Ni}_{1/3}\text{Mn}_{2/3}\text{O}_2$, shows a theoretical capacity of about 178 mAh/g when it fully utilizes Ni²⁺/3⁺/4⁺ redox. Also, the compound undergoes a phase transition from P2 to O2, however, it suffers from particle rupture due to the large volume change of about 23 %. However, the authors did not test the material over 4.15 V that starts to show another voltage plateau. As demonstrated in operando XRD, the material in this work presents a single-phase reaction, so that it was possible to have good electrode performance. Earlier work by S. Meng et al. present good cyclability when $\text{Na}_{2/3}\text{Ni}_{1/3}\text{Mn}_{2/3}\text{O}_2$ compound was tested below 4 V. From the above point of view, I have several comments.

1. Diffusivity shown in Figure 3g. For me, the P2- $\text{Na}_{0.78}\text{Ni}_{0.31}\text{Mn}_{0.67}\text{Nb}_{0.02}\text{O}_2$ seems to have a good diffusivity particularly at -40 oC. Did the author check LSV of the electrolyte at -40 oC that the potential window did not show oxidation over 4 V vs Na⁺/Na? After mixing with NaPF₆ salt, I wonder that the electrolyte did not show gelation or increase in viscosity at such a low temperature. Detailed data should be supplemented.
2. The GITT data shown in Figure 3g indicated that the GITT made at -40 oC exhibited lower IR drop than that of the data measured at RT. Detailed data should be provided.

3. Since temperature property seemed to be the main issue of this work, I recommend to test the material at 60 °C.
4. Rietveld refinement of XRD data. It is quite interesting to see that the occupation of Na edge site is rather higher than values reported in literature. What is the reason for that in structural aspect?
5. What is the motivation of the use of Nb element and limit it to 2 % in TM layers?
6. Figure 1a and b. Both compounds showed unusually high XRD intensity for 104, 100, and 110 planes. Detailed explanation should be made to help readers' understanding.
7. In consideration of ionic radius of Nb⁵⁺ (0.64 Å) with coordination number, 6, which is lower than Ni²⁺ (0.69 Å) and Mn³⁺ (0.645 Å low spin) but higher than Mn⁴⁺ (0.53 Å), it is not understandable why the resulting c-axis parameter increases by introduction of Nb⁵⁺.
8. TEM (Figure 2d). What is the reason for the formation of spinel-like phase on the surface? Indeed, Na-containing TM oxides does not form spinel phase due to the large ionic size of Na⁺. Moreover, the authors should check the residual sodium content of the compound. Since surface residues such as NaOH and/or Na₂CO₃ can be readily formed once the active materials are exposed in air, such surface residues may affect the formation of air-formed surface on the active materials. Seen from Figure 2i and j, it is more likely that the Nb-rich layer was formed on the surface of P2-Na_{0.78}Ni_{0.31}Mn_{0.67}Nb_{0.02}O₂.
9. Full cell. Unfortunately, I cannot see any information of electrode conditions for half- and full cells. The authors even did not describe the loading density or loading mass of active material for both cell tests. More seriously, data shown in Figure S9 is not the data of P2-Na_{0.78}Ni_{0.31}Mn_{0.67}Nb_{0.02}O₂//hard carbon. The feature is the cell using Na metal, because the voltage profile is the same as the data shown in Figure 3. The use of hard carbon should show sloppy rise and decay of voltage profile on the early of charge and the end of discharge, respectively. Furthermore, the irreversibility of hard carbon should induce small capacity of cathode, in this case it is estimated that the cathode may have the capacity of about 70 mAh/g. Notwithstanding, the delivered capacity was the same as the data shown in the half cells.
10. XPS presents only surface information. Detailed study to elucidate the variation of average oxidation state of TMs should be made by means of XANES.

Due to the above-mentioned unclarity, I do not recommend this work to be published in this leading journal.

Response to reviewer's comments:

Reviewer #1

The manuscript presents an idea of Nb-doped well-known Na-Mn-Ni-O system and will be helpful for researchers to select electrode materials for Sodium-ion batteries. The presented capacity of NaMNNb is relatively low, taking into account other layered oxides like Na-Mn-Fe-O or Na-Co-Fe-O systems. However, the material demonstrates high cycling stability.

Below I listed my comments. I recommend a major revision.

Response: Thank you for your positive and valuable comments. The following concerns have been well addressed accordingly and the revised manuscript is highly expected to be suitable for publishing in *Nature Communications*.

1 The authors indicated that nickel participates in the redox reactions during cell operation. How about Mn and Nb? Are these elements electrochemically inactive?

Response: Thank you for your valuable comments. For most P2-Manganese-based layered oxide, Mn mainly undergoes redox reactions of $\text{Mn}^{3+}/\text{Mn}^{4+}$ at 1.5-2.2 V. However, the structural stability will deteriorate due to the Jahn-Teller effect brought by Mn^{3+} , leading to inferior cycling performance. Therefore, we limit the discharge voltage at 2.4 V, so that only nickel participates in the redox reactions to provide capacity in our experimental range. Nb^{5+} is electrochemically inactive because of its insulating character without electrons in a conduction band ($4d^0$ configuration for Nb^{5+}), which is confirmed by *ex-situ* XPS in Supplementary Fig. 13.

2 How can you explain such low capacity, comparing with other well-known sodium layered oxides, e.g. $\text{Na}_{0.67}\text{Mn}_{0.5}\text{Fe}_{0.5}\text{O}_2$?

Response: Thank you for your comments. The low specific capacity in this article is due to the high cut-off voltage we used (~2.4 V). In fact, our as prepared P2-NaMNNb also exhibits a high specific capacity of 182.1 mAh g^{-1} in the voltage region of 1.5-4.15 V, which remains 88.8 mAh g^{-1} at 30 C, which is superior to that of other cathode in the wide voltage range. Besides, This result is now added to the supporting information (Supplementary Fig. 14 (Fig. R1), Supplementary Table 4 (Table. R1)). As contrast, $\text{Na}_{0.67}\text{Mn}_{0.5}\text{Fe}_{0.5}\text{O}_2$ exhibits ≈ 170 mAh g^{-1} in the voltage of 1.5-4.2 V, which is lower than our material under the same potential window.^{r1}

The reason why we choose such high cut-off voltage is as follows: i) the low redox potential of $\text{Mn}^{3+}/\text{Mn}^{4+}$ (1.5-2.2 V) would low down the working voltage of the positive electrode, which is not desired for enhancing the energy density in full-cell applications; ii) to avoid the Jahn-Teller effect cause by $\text{Mn}^{3+}/\text{Mn}^{4+}$ redox reaction at 1.5-2.2 V and improve the long-term cycling stability and rate capability. Generally, the specific capacity of Na_xMnO_2 based

material in literatures are around 80 mAh g^{-1} within $2.4\text{--}4.0 \text{ V}$,¹²⁻⁴ which is inferior to our work. In this work, by limiting the voltage region between $2.4\text{--}4.15 \text{ V}$, we can make full use of the capacity of high voltage platform ($3.2\text{--}3.6 \text{ V}$) to equip the full cell with hard carbon. The full cell demonstrates a high energy density of 216 Wh kg^{-1} , and extremely high power density of 6.6 kW kg^{-1} . As contrast, the as mentioned $\text{Na}_{0.67}\text{Mn}_{0.5}\text{Fe}_{0.5}\text{O}_2$ can only display a cycling stability of smaller than 100 cycles, and a poor rate performance of 3 C.

The electrochemical performance of P2-NaMNNb in the potential window of $1.5\text{--}4.15 \text{ V}$ is added to the supporting information (Supplementary Fig. 14 (Fig. R1)), and a comprehensive comparison of the electrochemical performance between our sample and previous work is listed in Supplementary Table 4 (Table R1):

Fig. R1 a Charge–discharge curves of P2-NaMNNb at 25 °C in 1.5-4.15 V. b The corresponding rate performance at 25 °C.

Supplementary Table 4 (Table R1) The comparison of electrochemical performance of various cathode oxide material.

Cathode	Potential window (V)	Rate performance (mAh g ⁻¹ / A g ⁻¹)	Cycling performance (% / cycling number)	Full cell cycling performance (% / cycling number)	Reference
Na _{2/3} Ni _{1/3} Mn _{2/3} O ₂	1.5-4.0	73.4	80.8% (500)	99 % (50)	r5
nanofibers		(3.5 A g ⁻¹)			
rhenanite-coated	2.5-4.3	62	74%		r6
Na _{2/3} [Ni _{1/3} Mn _{2/3}]O ₂		(2 A g ⁻¹)	(200)		
NaPO ₃ -coated	1.5-4.3	115	80% (50)	73% (300)	r7
Na _{2/3} [Ni _{1/3} Mn _{2/3}]O ₂		(1.12 A g ⁻¹)			
Na _{0.78} Cu _{0.27} Zn _{0.06} Mn _{0.67} O ₂	2.5-4.1	73 (5C)	85% (200)		r8
Na _{0.7} Mn _{0.6} Ni _{0.2} Mg _{0.2} O ₂	2.5-4.2	60	79% (1000)		r9
		(4.3 A g ⁻¹)			
Na _{0.67} Al _{0.1} Mn _{0.9} O ₂	2-4	74	86% (100)		r10
		(4.8 A g ⁻¹)			
Na _{0.67} Mn _{0.5} Fe _{0.47} Al _{0.03} O ₂	1.5-4.2	67	74.7% (50)		r1
		(0.4 A g ⁻¹)			
Na _{0.5} Ni _{0.1} Co _{0.15} Mn _{0.65} Mg _{0.1} O ₂	1.5-4.0	97.5	80.6% (200)	80.1% (200)	r11
		(0.8 A g ⁻¹)			
Na _{2/3} Ni _{1/3} Mn _{2/3} O _{1.95} F _{0.05}	2-4	86.4	89% (400)		r12
		(1.7 A g ⁻¹)			
Na _{0.85} Li _{0.12} Ni _{0.22} Mn _{0.66} O ₂	2-4.3	79.3	85.4% (500)	85.6% (100)	r13
		(4.2 A g ⁻¹)			
Na _{0.67} [Mn _{0.61} Ni _{0.28} Sb _{0.11}]O ₂	1.8-4.2	82	86% (200)		r14
		(1.4 A g ⁻¹)			
Na _{0.76} Ca _{0.05} [Ni _{0.23} □ _{0.08} Mn _{0.69}]O ₂	2.0-4.3	74.6	75.3% (200)	80.4% (100)	r15

		(2.4 A g ⁻¹)			
Na _{2/3} Ni _{0.25} Mg _{0.083} Mn _{0.55} Ti _{0.117} O ₂	3.0–4.4	84	87% (100)	88% (100)	r16
		(0.34 A g ⁻¹)			
Na _{0.78} Ni _{0.31} Mn _{0.67} Nb _{0.02} O ₂	2.4-4.15	65.8	76%	81.4% (400)	Our work
		(9.4 A g ⁻¹)	(1800,-40 °C)		

3 In Fig.3b there is lack of capacity value for C-rate 20C - 50C for NaMN. Is this because NaMN did not work in such high current range?

Response: Thank you for your valuable suggestions. Yes, the specific capacity of P2-NaMN for C-rate at 20 ~50 C is very low, about 40 mAh/g for C-rate of 20 C. This is because the P2-NaMN will undergo pulverization and corresponding transition metal dissolution under high current density, which results in the degradation of the structure. The structure instability will lead to the severe voltage polarization and deformed charge-discharge curve with the chemical reaction platform disappearing, resulting in insufficient electrochemical performance. As contrast, the Nb doping of P2-NaMNNb, could significantly improve the overall conductivity of the material, suppress the diffusion energy barrier of Na⁺, and meanwhile the Nb-induced surface-preconstruction would build up an electrochemical corrosion-resistant layer in atomic scale, which could prevent H₂O and solvent molecule enter the crystal lattice, suppress the structural distortion and enable a stable CEI film, which could prohibit the loss of transition metals. Therefore, this work demonstrates an efficient way to improve the structural stability and rate performance of Ni-Mn based oxides.

4 The authors showed only one charge/discharge curve for every current rate for the 25 °C. How about the first cycle, especially the charging process? Does it possess abnormal character or is it similar to these shown in Fig. 3c?

Response: Thank you for your valuable suggestion.

The charging and discharging curves of the first two cycles are now added to *Supplementary information* (Supplementary Fig. 10 (Fig. R2)). It can be seen the curves are similar to those in Fig.3c. The specific capacity is a little bit higher for the first cycle for the relative lower open circuit voltage than that of second cycle., which is common for the layer oxide of battery.²

Supplementary Fig. 10 (Fig. R2). The first two charge–discharge curves of P2-NaMNNb a, 25 °C and b, -40 °C.

5. Why did the authors mention the 1C value in the bracket? This value is multiplication of 96.6 mAh/g and does not mean the actual value obtained by the cell. Or maybe this value is theoretical value calculated for the electrode mass?

An explanation is necessary.

Response: Thank you for your valuable question.

C rate is a very common method to evaluate the rate performance of the battery materials. 1C = 184 mAh g⁻¹ is set as the theoretical specific capacity, calculated from according to the following formula. corresponding to 0.75 mol Na⁺ extracting from the structure.

$$C_0 = 26.8n \frac{m_0}{M}$$

Where C_0 is the theoretical specific capacity of P2-NaMNNb, n is the number of electrons transfer in electrode reaction. M is the molar mass of active material (106.8 g/mol). m_0 is the total mass of the active material reacted completely. A reference is added to this part: (1 C = 184 mAh g⁻¹)¹⁷.

6. The authors conducted the in-situ XRD measurement to have an insight into structural evolution of P2-NaMNNb cathode material during cell operation. But what was the base for identifying some peaks (e.g. 16.4 2theta) as P2' phase? P2-type structures are known that can transform to other types, like O2, OP4 etc. Did you evaluate it by Rietveld refinement? In the manuscript, as well as in Supplementary information, I did not find suitable explanation. You should also add to the text the JCPD number of the phases you used.

Response: Thank you for your helpful suggestion. P2-type structures are known that can transform to other types, like O2, OP4 etc. the O2 or OP4 phase that P2 transformed is an severe distortion (23% volume expansion) and often occur during charging process at high voltage with the diffraction peak of (002) shift to higher angle around 20°. The diffraction pattern at 16.4° of P2' phase were indexed to orthorhombic lattice with the space group *Cmcm* (JCPDF no.

27-752). This phase is a distorted hexagonal P2 phase with negligible variation of β from 90° to 90.68° and often appeared under 2.7 V. The P2' phase our reported is also fits with diffraction angle of the previous literature.^{r9,r10,r18}

We have made corresponding corrections in the maintext, which is highlighted in yellow color.

7. In the experimental part in “Electrochemical measurement”: there is a lack of chemical substances manufacturers. Moreover, what kind of test cell cases did you use?

Response: Thank you for your valuable suggestion. We have made corresponding corrections in the “Electrochemical measurement” part of the revised manuscript, which is highlighted in yellow color.

“The 2032 coin cells were prepared in an argon-filled glove box. All chemical material were provided by Guangdong Canrd New Energy Technology Co.,Ltd. (China)” .

8. Please, check your manuscript, because some typos and grammar mistakes are present.

Response: Thanks for your kind reminder. According to your suggestion, we have double checked the manuscript and revised it in detail.

Reviewer #2: Shi et al. present an interesting property of P2-Na_{0.78}Ni_{0.31}Mn_{0.67}Nb_{0.02}O₂ layer compound that was active even at -40 °C, using an electrolyte of 1M NaPF₆ in diglyme in Na half- and full cells. A similar compound, Na_{2/3}Ni_{1/3}Mn_{2/3}O₂, shows a theoretical capacity of about 178 mAh/g when it fully utilizes Ni^{2+/3+/4+} redox. Also, the compound undergoes a phase transition from P2 to O2, however, it suffers from particle rupture due to the large volume change of about 23 %. However, the authors did not test the material over 4.15 V that starts to show another voltage plateau. As demonstrated in operando XRD, the material in this work presents a single-phase reaction, so that it was possible to have good electrode performance. Earlier work by S. Meng et al. present good cyclability when Na_{2/3}Ni_{1/3}Mn_{2/3}O₂ compound was tested below 4 V. From the above point of view, I have several comments.

Response: Thank you very much for your arduous work and comments. We have studied the comments carefully and the following revision concerns have been well addressed accordingly.

It is true that Na_{2/3}Ni_{1/3}Mn_{2/3}O₂, shows a theoretical capacity of about 178 mAh/g when it fully utilizes Ni^{2+/3+/4+} redox, which undergoes P2 to O2 phase transitions above 4.15V. To date, lots of efforts have been taken to suppress the P2-O2 phase transitions of the P2 phase material, but the long-term stability remains unsatisfied. Therefore, the motivation of this work is to maximize the capacity contribution of the voltage pleatue at ~3.6 and ~3.2 V, while avoid using the capacity above 4.15 V. The as prepared P2-NaMNNb demonstrates a high specific capacity of 96.6 mAh g⁻¹ in the voltage range of 2.4 - 4.15 V at 0.5 C (1 C = 184 mAh g⁻¹), which is only 84.4 mAh g⁻¹ for P2-NaMN.

Besides an ultrahigh rate performance of P2-NaMNNb (50 C) is achieved. This value is 100 folds higher than that of the as mentioned literature by Meng et al. (Phys. Chem. Chem. Phys., 2013, 15, 3304), which only demonstrates a rate performance of 2C, and a specific capacity of 83.3 mAh g⁻¹ at 0.5 C. The cycling performance of their sample is very poor, which is only 50 cycles for pure Na_{2/3}Ni_{1/3}Mn_{2/3}O₂ (Phys. Chem. Chem. Phys., 2013, 15, 3304), and 100 cycles (55% retention) for Al₂O₃ coated Na_{2/3}Ni_{1/3}Mn_{2/3}O₂ (ACS Appl. Mater. Interfaces 2017, 9, 26518–26530).

It is obvious that, such a design can not only maintain the high average voltage, but also avoid the irreversible phase transition above 4.15 V. Therefore, our work provides a new approach by maximizing the capacity in a proper voltage range, to achieve the high voltage plateau, high rate performance, and high stability simultaneously, which enables a high energy density and power density for the full-cell. This work not only provides a thorough understanding of the high valence element (Nb⁵⁺) doping and surface pre-construction for the classic Mn-Ni-oxides based P2 phase, but also demonstrates a practical way for the grid-scale applications of P2 materials.

We appreciate your valuable suggestions to improve the quality of our manuscript and the revised manuscript is highly expected to be suitable for publishing in *Nature Communications*.

1. Diffusivity shown in Figure 3g. For me, the P2-Na_{0.78}Ni_{0.31}Mn_{0.67}Nb_{0.02}O₂ seems to have a good diffusivity particularly at -40 °C. Did the author check LSV of the electrolyte at -40 °C that the potential window did not show oxidation over 4 V vs Na⁺/Na? After mixing with NaPF₆ salt, I wonder that the electrolyte did not show gelation or increase in viscosity at such a low temperature. Detailed data should be supplemented.

Response: Thank you for your valuable suggestion. We have checked LSV of the electrolyte at -40 °C and investigated the temperature durability of the electrolyte. We have added the corresponding description to the maintext as well as the supporting information (Supplementary Fig. 19-20 (Fig. R3-R4)) and marked it yellow in our revised manuscript.

Supplementary Fig. 19 (Fig. R3) Optical photos of electrolyte (1 mol NaPF₆ in Diglyme) put in an environment of -40 °C a) pristine state, b) 1 day, c) 3 days.

Supplementary Fig. 20 (Fig. R4) LSV curves of 1 mol NaPF₆ in Diglyme based on Na-Al cells at -40 °C.

2. The GITT data shown in Figure 3g indicated that the GITT made at -40 °C exhibited lower IR drop than that of the data measured at RT. Detailed data should be provided?

Response: Thank you for your comments. Polarization in GITT curves comprises voltage polarization and ohmic polarization (IR drop), as depicted by the enlarged image inset in Supplementary Fig. 16, which have been reported by previous work.^{r3,r20,r21} The obvious voltage polarization of P2-NaMNNb at 3.5 V and 3.8 V at 25 °C should be attributed to the parameter set up during the GITT test. To make the data comparable, we chose the same test step at RT and LT. But the capacity at 25°C is higher than -40°C, no capacity contribution within this voltage range, the voltage polarization will increase once charging stops in this area. This is also consistent with the drop of diffusion coefficient in 15 h and 30 h in Fig 3h. However, under the same test step, due to the difference of capacity between at 25 °C and -40 °C, the GITT points taken are varied. The above two position of GITT points at 25 °C are obtained without contribution to the capacity, which lead to obvious voltage polarization but negligible IR drop. Besides, we have double checked the result, as shown in supplementary Fig 16, the IR-drop of P2-NaMNNb at 25 °C is lower than that at -40 °C. The corresponding correction has been made in the supplementary information (Supplementary Fig. 16 (Fig. R5)), and highlighted in yellow color.

Supplementary Fig. 16 (Fig. R5) Calculated voltage polarization and IR-drop from GITT data during charge process.

3. Since temperature property seemed to be the main issue of this work, I recommend to test the material at 60 °C.

Response: Thank you for your suggestions.

We have tested the material at 60 °C according to your comment. Unfortunately, the charge curve display an abnormal long platform at 3.5 V, which indicates the electrolyte goes through obviously oxidation and decomposition at high temperature (Fig. R6a). It should be noted that, the main innovation of our work lies in the electrode material, and to find temperature-durable electrolyte is not the main focus of this work. Instead, we have measured the electrochemical performance of P2-NaMNNb at 40 °C using the same electrolyte (Fig R6b). The capacity retention is 80.4% after 500 cycles in the current rate of 2 C, which is close to that at 25 °C, indicating an eminent durability in relatively high temperature (Fig. R6c).

Fig. R6. The charging and discharging curves of P2-NaMNNb at a 60 °C, b 40 °C. c Long-term cycling stability in a rate of 2 C at 40 °C.

4. Rietveld refinement of XRD data. It is quite interesting to see that the occupation of Na edge site is rather higher than values reported in literature. What is the reason for that in structural aspect?

Response: Thank you for this comment. In P2-type oxide compounds, Na ions occupy two kinds of trigonal prismatic sites: One is Na_f 2b site shares two faces with the lower and upper octahedral TMO₆, which is expected to be less stable than that of Na_e 2d site sharing two edges with six octahedral TMO₆ because of the higher occupation energy. The ionic radius of Nb⁵⁺ (0.64 Å) with coordination number, 6, which is lower than Ni²⁺ (0.69 Å) and Mn³⁺ (0.645 Å low spin) but higher than Mn⁴⁺ (0.53 Å). Nb⁵⁺ with similar ionic radii to Mn and Ni ions, but with more electron holes to regulate the ratio of Na_e and Na_f occupancy. The Nb⁵⁺ substitution enlarged Na-O bond but shortened TM-O bond (Figure 5d in revised manuscript), indicating that a stronger Nb⁵⁺-Na_f columbic repulsion due to special sites of Na_f below or above of Nb⁵⁺. The intense columbic repulsion of Nb⁵⁺-Na_f could prompt some Na_f ions to move to the Na_e sites, leading to less occupancy of Na_f sites. Therefore, the occupation of Na edge site is obviously higher

than that of Na face site.

5. What is the motivation of the use of Nb element and limit it to 2 % in TM layers?

Response: Thank you for your helpful question.

We chose Nb as doping element due to the following considerations: 1) Nb^{5+} has been introduced to in the nickel-rich layered oxide cathodes for lithium-ion batteries to stabilize structure, thus inhibiting potential decaying during cycling process.^{r22} 2) The bonding energy of Nb-O (753 kJ mol^{-1}) > Mn-O (402 kJ mol^{-1}) > Ni-O ($391.6 \text{ kJ mol}^{-1}$) > Co-O (368 kJ mol^{-1}),^{r23} which seem the structure stability could be enhanced by the stronger bond energy of Nb-O. Considering the information above, we choose to introduce Nb^{5+} into high Na-content cathode oxide to investigate its electrochemical and theoretical behavior in sodium ion battery.

The successful doping of elements is both determined by the size effect and the charge effect. Since the valance of Nb is +5, the doping amount of Nb might be limited by the charge effect. We also tried doping with different amounts of Nb, and found that when the Nb content is greater than 2%, an impurity phase (Na_3NbO_4) will be precipitated: Fig. R7 showed the XRD patterns of samples doped with varied amount of Nb (Nb-0.05 and Nb-0.07), it is found that additional peaks appeared at 33.5° and 39° , which belong to Na_3NbO_4 phase (JCPDF: No 22-1391). The intensity of (200) plane of Na_3NbO_4 increases with the amount of Nb increase.

Fig. R7 Comparison of the XRD diffraction pattern and corresponding electrochemical curve of P2-NaMNNb, Nb-0.05, Nb-0.07.

6. Figure 1a and b. Both compounds showed unusually high XRD intensity for 104, 100, and 110 planes. Detailed explanation should be made to help readers' understanding.

Response: Thank you for your valuable comments. Both compounds showed preferred orientation of XRD intensity for (104), (100), and (110) planes. This could be attributed to the synthesis method. As reported by previous

work, P2-type materials calcined by carbonate precursors through co-precipitating methods usually exhibit preferred orientation of XRD intensity for (104), (100), and (110) planes,^{r24,r25} which is different from the preferred orientation for (102) and (106) plane through a solid-state reaction with oxide precursors.^{r12,r26} We have revised in manuscript accordingly:

7. In consideration of ionic radius of Nb⁵⁺ (0.64 Å) with coordination number, 6, which is lower than Ni²⁺ (0.69 Å) and Mn³⁺ (0.645 Å low spin) but higher than Mn⁴⁺ (0.53 Å), it is not understandable why the resulting *c*-axis parameter increases by introduction of Nb⁵⁺.

Response: Thanks for your valuable questions. The parameters of the crystal structure after doping is co-determined by the size effect and the charge effect of doping ion. Many transition metal ion were chosen to dope in the crystal structure regulating the *c*-axis. For instance, Al³⁺ (0.54 Å) doped into the Na_{0.67}MnO₂ could increase the *c*-axis from 11.208 Å to 11.263 Å.^{r10} Jiao et al. developed a Na_{0.85}Li_{0.12}Ni_{0.22}Mn_{0.66}O₂ by utilizing Li⁺ (0.76 Å) to inhibit irreversible P2-O2 phase transformation, and decrease the *c*-axis from 11.145 Å to 11.010 Å.^{r13} Zhou et al. synthesized Na_{0.66}[Mn_{0.61}Ni_{0.28}Sb_{0.11}]O₂ with introduction of Sb⁵⁺ (0.6 Å) to tune Na_e and Na_f site, increasing the *c*-axis from 11.179 Å to 11.241 Å.^{r14} Hu et al. also mentioned that doping transition metal ion with high oxidation state could increase the *c*-axis.^{r4} Therefore, although the radius Nb⁵⁺ is similar to that of Ni²⁺ and Mn³⁺, its high valance (5+) would cause things different.

8. TEM (Figure 2d). What is the reason for the formation of spinel-like phase on the surface? Indeed, Na-containing TM oxides does not form spinel phase due to the large ionic size of Na⁺. Moreover, the authors should check the residual sodium content of the compound. Since surface residues such as NaOH and/or Na₂CO₃ can be readily formed once the active materials are exposed in air, such surface residues may affect the formation of air-formed surface on the active materials. Seen from Figure 2i and j, it is more likely that the Nb-rich layer was formed on the surface of P2-Na_{0.78}Ni_{0.31}Mn_{0.67}Nb_{0.02}O₂.

Response: Thanks to the reviewer for asking this important question.

To further understand how the Nb-rich spinel-like phase formed on the surface, theoretical calculations and corresponding experimental characterizations were carried out. The formation energy depicted that Nb is prone to replace the Ni site in the bulk phase (-4.67 eV) compared to -4.01 eV and -2.29 eV of Mn and Na site respectively, and Na site in the surface phase (-5.59 eV). Moreover, It is noteworthy that the formation energy is -4.81 eV when all Na sites on the surface are substituted by Nb, remaining lower than that of Ni site (-4.67 eV) in the bulk phase (Supplementary Fig. 7 (Fig. R8), Supplementary Table 5 (Table. R2)). Therefore, in the synthesis process, most

surface Na atoms could be readily replaced by Nb, while a few amount of Nb will be dopd to the Ni site in the bulk phase. Furthermore, when Na⁺ replaced by Nb⁵⁺ on the surface, the charge density around O_{2p} has varied due to the stronger bonding energy of Nb-O than Na-O. The electrostatic interaction between TM and O will alter as the amount of Nb increase, resulting in the transformation process from bulk (P63/mmc) to surface (Fd3m/Fm3m). Such a structure could actually enhance the air-stability of the material. Moreover, cathode–electrolyte interface (CEI) of P2-NaMNNb (2-5 nm) is thinner than that of P2-NaMN (18.2 nm), which also reveal that the pre-constructed layer is a critical factor to the formation of CEI and eminent rate performance (Fig. 4d-e (Fig R9d-e), Supplementary Fig. 23 (Fig. R10)). In addition, according to our TEM and XRD results, NaOH and/or Na₂CO₃ are not found on the surface. The corresponding explanation is added to the maintext and supporting information:

Supplementary Fig. 7 (Fig. R8) Schematic diagram of the substitution of Nb⁵⁺ in surface of (-103) plane. **a** an original plane, **b** replace to single Na⁺, **c** replace all Na⁺. **d** The comparison of formation energy of different substitution site of Nb (ANa represent all Na are replaced by Nb in the surface layer).

Supplementary Table 5 (Table. R2) The formation energy of the different substitution site of Nb.

Formation energy (eV)			
original ion	Ni	Na	Mn
bulk	-4.67	-4.01	-2.29
surface	-4.88	-5.59	-0.27

Fig 4 (Fig. R9) a *In situ* XRD patterns corresponding to the charge and discharge curves between 2.4-4.15 V. b Comparison of (002) diffraction patterns of P2-NaMNNb and P2-NaMN after 30 cycles at 0.5 C. The HAADF-STEM image c, and ABF-STEM d and e images of the P2-NaMNNb after 30 cycles. f Schematic of the protective effect of surface pre-reconstruction on the bulk structure.

Supplementary Fig. 23 (Fig. R10) The HRTEM image of **a** P2-NaMNNb and **b** P2-NaMN after 30 cycles.

9. Full cell. Unfortunately, I cannot see any information of electrode conditions for half- and full cells. The authors even did not describe the loading density or loading mass of active material for both cell tests. More seriously, data shown in Figure S9 is not the data of P2-Na_{0.78}Ni_{0.31}Mn_{0.67}Nb_{0.02}O₂//hard carbon. The feature is the cell using Na metal, because the voltage profile is the same as the data shown in Figure 3. The use of hard carbon should show sloppy rise and decay of voltage profile on the early of charge and the end of discharge, respectively. Furthermore, the irreversibility of hard carbon should induce small capacity of cathode, in this case it is estimated that the cathode may have the capacity of about 70 mAh/g. Notwithstanding, the delivered capacity was the same as the data shown in the half cells.

Response: Thank the reviewer for this question. Data shown in Figure S9 is the electrochemical curve of half cell of P2-Na_{0.78}Ni_{0.31}Mn_{0.67}Nb_{0.02}O₂//Na. The electrochemical curve data of full cell of P2-Na_{0.78}Ni_{0.31}Mn_{0.67}Nb_{0.02}O₂//hard carbon is Figure S19 in original manuscript. The voltage platform for charging and discharging curve of full cell is slightly lower than that of half cell, which because of the decay of voltage profile caused by the matching of hard carbon and P2-NaMNNb (Supplementary Fig. 28 (Fig. R11), Fig. R12). However, the electrochemical curve of full cell is still similar to that of half cell, which should be attributed to the sufficient presodiation of hard carbon and suitable mass ratio between negative electrode and positive electrode (N/P was around 1.6-1.8), as well as appropriate potential window (2.3–4.14 V). The similar electrochemical curve of between full cell and half cell is widely reported by previous researchers.^{r4,r5,r27,28} Hu et al. constructed a full cell of Na_{45/54}Ni_{16/54}Mn_{34/54}Li_{4/54}O₂//hard carbon, which deliver the reversible capacity of 100 mAh g⁻¹ between 1.5 and 4.0 V, compared to 100 mAh g⁻¹ of half cell in the voltage of 2-4 V.^{r4} Fan et al. assemble the full cell of Na_{0.67}Ni_{0.33}Mn_{0.67}O₂//hard carbon, which exhibit 160 mAh g⁻¹ within the 1.2-3.8 V, close to the specific capacity of the half cell (160 mAh g⁻¹).^{r5} Myung et al. developed

$\text{Na}_{0.9}[\text{Ni}_{0.1}\text{Fe}_{0.1}\text{Mn}_{0.8}]\text{O}_2$ /hard carbon with 210 mAh g^{-1} in 1.4-4.2 V, which is almost the same to the capacity of half cells.¹²⁷ So the delivered capacity of the full cell was close to that of the half cell can be accepted if the potential window and ratio of N/P are suitable.

Fig. R11 Galvanostatic charge/discharge curves of the hard carbon electrode.

Fig. R12 The comparison of electrochemical curve of half-cell and full-cell.

10. XPS presents only surface information. Detailed study to elucidate the variation of average oxidation state of TMs should be made by means of XANES.

Response: Thanks for the reminder of the reviewer. The valence states of these elements in these reactions are very unstable. We are very sorry for we have no conditions to do *in-situ* XANES due to the COVID-19. We have employed the oxidation state of TMs by means of XANES and revised in supplementary information (Supplementary Fig. 12 (Fig. R13)). We also investigate the variation of average oxidation state of TMs after cycling process (Fig R14). It is regrettable that the sample reacts with the air without suitable protective measures, and the data has no reference value. We also do the EELS to verify that the difference of valence of Mn and Ni between surface and bulk. The

results deliver that the valence state of Mn in the pre-constructed layer is lower than that in bulk phase (the mixture of Mn^{3+} and Mn^{4+}), which could be attributed to the enrichment of high oxidation state of Nb^{5+} at the surface, reducing Mn^{4+} to Mn^{3+} to satisfy the conservation of charge. Besides, the shape and position of K-edge of O, L-edge of Mn and Ni after cycle are also in alignment with that of pristine state (Supplementary Fig. 24 (Fig. R15), Supplementary Fig. 25 (Fig. R16)), indicating such pre-constructed layer exhibit splendid electrochemical corrosion durability and stability.

Fig. R13 XANES of **a** Mn and **b** Ni K-edge of P2-NaMNNb and P2-NaMN as well as standard metal oxide references.

Fig. R14 The ex-situ XANES of P2-NaMNNb with different cutoff voltage **a** Mn, **b** Ni.

Fig. R15 The image of EELS area of P2-NaMNNb **a**. The EELS spectra of **b**, Mn L-edge (Mn L_{2,3}; **c**, Ni L-edge ; **d**, O K-edge.

Fig. R16 **a** The image of EELS area of P2-NaMNNb after 30 cycles. The EELS spectra of **b**, Mn L-edge (Mn L_{2,3}; **c**, Ni L-edge ; **d**, O K-edge.

Reference

- r1. Wang, H., et al. Different Effects of Al Substitution for Mn or Fe on the Structure and Electrochemical Properties of $\text{Na}_{0.67}\text{Mn}_{0.5}\text{Fe}_{0.5}\text{O}_2$ as a Sodium Ion Battery Cathode Material. *Inorg. Chem.* **57**, 5249-5257 (2018).
- r2. Wang, P.-F., et al. $\text{Na}^{+}/\text{vacancy}$ disordering promises high-rate Na-ion batteries. *Science Advances* **4**, eaar6018 (2018).
- r3. Xiao, Y., et al. A Stable Layered Oxide Cathode Material for High-Performance Sodium-Ion Battery. *Adv. Energy Mater.* **9**, 1803978 (2019).
- r4. Zhao, C., et al. Revealing High Na-Content P2-Type Layered Oxides as Advanced Sodium-Ion Cathodes. *J. Am. Chem. Soc.* **142**, 5742-5750 (2020).
- r5. Liu, Y., et al. Hierarchical Engineering of Porous P2- $\text{Na}_{2/3}\text{Ni}_{1/3}\text{Mn}_{2/3}\text{O}_2$ Nanofibers Assembled by Nanoparticles Enables Superior Sodium-Ion Storage Cathodes. *Adv. Funct. Mater.* **30**, 1907837 (2020).
- r6. Jo, C.-H., et al. Bioinspired Surface Layer for the Cathode Material of High-Energy-Density Sodium-Ion Batteries. *Adv. Energy Mater.* **8**, 1702942 (2018).
- r7. Jo, J. H., et al. Sodium-Ion Batteries: Building Effective Layered Cathode Materials with Long-Term Cycling by Modifying the Surface via Sodium Phosphate. *Adv. Funct. Mater.* **28**, 1705968 (2018).
- r8. Yan, Z., et al. A Hydrostable Cathode Material Based on the Layered P2@P3 Composite that Shows Redox Behavior for Copper in High-Rate and Long-Cycling Sodium-Ion Batteries. *Angew. Chem. Int. Ed.* **58**, 1412-1416 (2019).
- r9. Wang, Q.-C., et al. Tuning P2-Structured Cathode Material by Na-Site Mg Substitution for Na-Ion Batteries. *J. Am. Chem. Soc.* **141**, 840-848 (2019).
- r10. Liu, X., et al. P2- $\text{Na}_{0.67}\text{Al}_x\text{Mn}_{1-x}\text{O}_2$: Cost-Effective, Stable and High-Rate Sodium Electrodes by Suppressing Phase Transitions and Enhancing Sodium Cation Mobility. *Angew. Chem. Int. Ed.* **58**, 18086-18095 (2019).
- r11. Zhu, Y.-F., et al. Manipulating Layered P2@P3 Integrated Spinel Structure Evolution for High-Performance Sodium-Ion Batteries. *Angew. Chem. Int. Ed.* **59**, 9299-9304 (2020).
- r12. Liu, K., et al. Insights into the Enhanced Cycle and Rate Performances of the F-Substituted P2-Type Oxide Cathodes for Sodium-Ion Batteries. *Adv. Energy Mater.* **10**, 2000135 (2020).
- r13. Jin, T., et al. Realizing Complete Solid-Solution Reaction in High Sodium Content P2-Type Cathode for High-Performance Sodium-Ion Batteries. *Angew. Chem. Int. Ed.* **59**, 14511-14516 (2020).
- r14. Wang, Q.-C., et al. Tuning Sodium Occupancy Sites in P2-Layered Cathode Material for Enhancing Electrochemical Performance. *Adv. Energy Mater.* **11**, 2003455 (2021).

-
- r15. Shen, Q., et al. Transition-Metal Vacancy Manufacturing and Sodium-Site Doping Enable a High-Performance Layered Oxide Cathode through Cationic and Anionic Redox Chemistry. *Adv. Funct. Mater.* **n/a**, 2106923
- r16. Huang, Y., et al. Vitalization of P2-Na_{2/3}Ni_{1/3}Mn_{2/3}O₂ at high-voltage cyclability via combined structural modulation for sodium-ion batteries. *Energy Storage Mater.* **29**, 182-189 (2020).
- r17. Jin, T., et al. Polyanion-type cathode materials for sodium-ion batteries. *Chem. Soc. Rev.* **49**, 2342-2377 (2020).
- r18. Paulsen, J. M. & Dahn, J. R. Studies of the layered manganese bronzes, Na_{2/3}[Mn_{1-x}Mx]O₂ with M=Co, Ni, Li, and Li_{2/3}[Mn_{1-x}Mx]O₂ prepared by ion-exchange. *Solid State Ionics* **126**, 3-24 (1999).
- r19. Hou, B.-H., et al. Self-Supporting, Flexible, Additive-Free, and Scalable Hard Carbon Paper Self-Interwoven by 1D Microbelts: Superb Room/Low-Temperature Sodium Storage and Working Mechanism. *Adv. Mater.* **31**, 1903125 (2019).
- r20. Shen, Z., Cao, L., Rahn, C. D. & Wang, C.-Y. Least Squares Galvanostatic Intermittent Titration Technique (LS-GITT) for Accurate Solid Phase Diffusivity Measurement. *J. Electrochem. Soc.* **160**, A1842-A1846 (2013).
- r21. Weppner, W. & Huggins, R. A. Determination of the Kinetic Parameters of Mixed-Conducting Electrodes and Application to the System Li₃Sb. *J. Electrochem. Soc.* **124**, 1569-1578 (1977).
- r22. Xin, F., et al. What is the Role of Nb in Nickel-Rich Layered Oxide Cathodes for Lithium-Ion Batteries? *Acs Energy Lett.* **6**, 1377-1382 (2021).
- r23. Linden, D. Handbook of batteries. *Fuel and Energy Abstracts* **36**, 265 (1995).
- r24. Liu, Z., et al. Ultralow Volume Change of P2-type Layered Oxide Cathode with Controlled Phase Transition by Regulating Distribution of Na⁺. *Angew. Chem. Int. Ed.* **n/a**,
- r25. Ma, C., et al. Exploring Oxygen Activity in the High Energy P2-Type Na_{0.78}Ni_{0.23}Mn_{0.69}O₂ Cathode Material for Na-Ion Batteries. *J. Am. Chem. Soc.* **139**, 4835-4845 (2017).
- r26. Wang, P.-F., et al. Suppressing the P2-O₂ Phase Transition of Na_{0.67}Mn_{0.67}Ni_{0.33}O₂ by Magnesium Substitution for Improved Sodium-Ion Batteries. *Angew. Chem. Int. Ed.* **55**, 7445-7449 (2016).
- r27. Choi, J. U., Jo, J. H., Park, Y. J., Lee, K.-S. & Myung, S.-T. Mn-Rich P2-Na_{0.67}[Ni_{0.1}Fe_{0.1}Mn_{0.8}]O₂ as High-Energy-Density and Long-Life Cathode Material for Sodium-Ion Batteries. *Adv. Energy Mater.* **10**, 2001346 (2020).
- r28. Park, Y. J., et al. A New Strategy to Build a High-Performance P2-Type Cathode Material through Titanium Doping for Sodium-Ion Batteries. *Adv. Funct. Mater.* **29**, 1901912 (2019).
- r29. Liu, S., et al. Li-Ti Cation Mixing Enhanced Structural and Performance Stability of Li-Rich Layered Oxide. *Adv. Energy Mater.* **9**, 1901530 (2019).

Reviewers' Comments:

Reviewer #2:

Remarks to the Author:

The authors responded nicely, except the full cells. It is quite difficult to understand how the half- and full cell capacities are same, although the authors have presodiated the hard carbon electrode. And the N/P ratio was about 1.6-1.8, which is very high for fabrication of full cell. When hard carbon is used as the anode, the curve shape is not similar to the shape obtained in the half cell with Na metal anode. The authors witnessed the sloppy decay of hard carbon anode in the range of 0.08-0.2 V in Fig. R11. The voltage profile of the full cell should have shown more curvy below 3.2 V due to the curvy shape in voltage profile of hard carbon anode in the voltage range.

More seriously, the capacity unbalance ($N/P = 1.6 - 1.8$) can cause a dramatic capacity fading of the full, while the capacity retention seemed to be fine for 400 cycles.

Note that the capacity shown in Fig. R12 reached about 100 mAh/g, however, the capacity shown in Figure 6b is lower than 90 mAh/g for 25 oC. And what I cannot understand is that the obtained capacity at -40 oC is higher than the capacity obtained at 25 oC.

The inconsistency of the cell performance in Fig. 6d and Fig. R12 is very confusing, so that I do not agree with publication of this work in this leading journal.

Manuscript Number: NCOMMS-21-32268A-Z

Title: Regulating Surface and Local Chemistry in High Na-content P2-type Cathode to Achieve Ultrahigh Power and Low Temperature Sodium Storage

Dear Editor and Reviewers,

Thank you very much for arduous work and valuable comments. We have studied the comments carefully and we are pleased to find the positive comments in the respects of the interest and performance improvement on low-temperature sodium ion batteries from both Reviewers. The main problem raised by the reviewers, including the voltage profile of half- and full-cell in this work and the specific capacity values, have been fully addressed in our revised manuscript. Here within the enclosure is our revised manuscript, which has been improved according to your suggestions.

Revised parts are highlighted with yellow color, and the point to point responses to the Reviewer's comments are listed below:

Comments from Reviewer#2:

Reviewer #2 (Remarks to the Author):

The authors responded nicely, except the full-cells. It is quite difficult to understand how the half- and full-cell capacities are same, although the authors have presodiated the hard carbon electrode. And the N/P ratio was about 1.6-1.8, which is very high for fabrication of full-cell. When hard carbon is used as the anode, the curve shape is not similar to the shape obtained in the half-cell with Na metal anode. The authors witnessed the sloppy decay of hard carbon anode in the range of 0.08-0.2 V in Fig. R11. The voltage profile of the full-cell should have shown more curvy below 3.2 V due to the curvy shape in voltage profile of hard carbon anode in the voltage range.

More seriously, the capacity unbalance ($N/P = 1.6 - 1.8$) can cause a dramatic capacity fading of the full, while the capacity retention seemed to be fine for 400 cycles.

Note that the capacity shown in Fig. R12 reached about 100 mAh/g, however, the capacity shown in Figure 6b is lower than 90 mAh/g for 25 °C. And what I cannot understand is that the obtained capacity at -40 °C is higher than the capacity obtained at 25 °C.

The inconsistency of the cell performance in Fig. 6d and Fig. R12 is very confusing, so that I do not agree with publication of this work in this leading journal.

Response:

(1) About the capacity value at -40 °C and 25 °C in Fig. R12 and Figure 6d

Answer: In fact, the capacity for the RT (25 °C) shown in Figure 6d is tested at 5 C, but the LT (at -40 °C) performance is tested at 0.5 C. The rate difference between the two samples is clearly labeled in the figures. In both Figure 6b and Figure R12, the capacity is about 100 mAh/g, which was calculated based on the mass of cathode materials.

(2) About “the capacity unbalance (N/P = 1.6 - 1.8) can cause a dramatic capacity fading of the full”

Answer: It is true if the specific capacity of the full-cell was calculated based on the total mass of both electrodes. However, the specific capacity in our previous version was based on the mass of cathode material, such a calculation way has been adopted in many literatures.^{R1-R5} We have now re-calculated the capacity based on the total mass of cathode and anode material in our revised manuscript, which is 60.4 mAh g⁻¹.

(3) About “The voltage profile of the full-cell should have shown more curvy below 3.2 V due to the curvy shape in voltage profile of hard carbon anode in the voltage range.”

Answer: To achieve a thorough understanding about the galvanostatic charge/discharge (GCD) curve shape of the full-cell, we have done a lot of experiments. We found that, the GCD curves of the full-cell can be affected by lots of factors, such as the ratio between the anode and cathode material (N/P capacity ratio), the pre-sodiation degree or

method, the testing conditions (e.g. the potential window etc.), as well as the property of the cathode materials etc.

① The effect on the full-cell galvanostatic charge/discharge (GCD) curve shape from various factors

Generally the ICE (initial coulombic efficiency) of the hard carbons cannot satisfy the full-cell construction, which demands a pre-sodiation of the hard carbons to makeup the capacity loss consumed by the SEI (solid electrode interface) formation in the first cycle. Several types of pre-sodiation have been developed^{R6}, including electrochemical pre-cycle (electrochemical pre-sodiation)^{R3,R7}, physical contact with Na (physical pre-sodiation)^{R8,R9}, chemical presodiation^{R10-R12}, or even self-presodiation^{R13}, etc. Among them, the electrochemical pre-sodiation and the physical contact method are most often used. We have summarized the full-cell performance with different pre-sodiation methods reported in literature (Table R1). It is noticed that, the shape of voltage profiles for the full-cells often show very similar shape with that of their half-cells after physical pre-sodiation (e.g. *Adv.Mater.* 2021, 2103210, *Adv. Funct. Mater.* 2019, 29, 1901912)^{R9,14} (see in Fig. R1a, b), which sometimes can also happen to those full-cell with electrochemical pre-sodiation method (e.g. *Nat. Commun* 12, 2256 (2021), *Nat. Commun* 12, 5267 (2021))^{R3,R7}.

Fig. R1 The galvanostatic charge/discharge curve of **a** half-cell, **b** full-cell from literature.^{R9}

We therefore took efforts to verify the effect of pre-sodiation method on the GCD curve shape (Fig R2-R6), and we have several findings: It is found that, with the same N/P ratio, the voltage profile from the electrochemical pre-

sodiation tends to be curvier than that with physical pre-sodiation (Fig. R2). On the other hand, when we vary the N/P ratio, the full-cell with less negative electrode tends to be curvier (Fig. R3).

Fig. R2 The comparison of galvanostatic charge/discharge curve of full-cell with different pre-sodiation method.

Fig. R3 The comparison of galvanostatic charge/discharge curve of full-cell with different ratio of N/P after: **a** physical pre-sodiation, **b** electrochemical pre-sodiation.

Such a phenomenon could be attributed to the capacity usage of the HCs. As well known, the typical GCD curve of the HCs shows a plateau region (< 0.1 V) and a slope region (> 0.1 V) (Fig. R4). When compose the full-cell, the

potential window and different N/P ratio would together decide the capacity contribution percentage of the anode and cathode. For example, in our work, the potential window of the full-cell is 1.85 V (4.15-2.3 V), which allows the overlap of the capacity contribution from anode (HCs: 0.01~2.0 V) and cathode (NNM: 2.4~4.15 V). In this case, the N/P ratio would be critical to determine how much capacity of the anode would be used in the full-cell. With high N/P ratio (>1.5, electrochemical pre-sodiation), only the capacity from the plateau region of HCs can be taken into use, which would lead to the unobvious GCD shape change as compared to that of the half-cell. But with low N/P (<1.5), the capacity from the slope region can also be used, whereby the GCD shows obvious curvy shape below 3.2 V (Fig. R3b). Such phenomenon has been witnessed in both pre-sodiation methods, but for physical pre-sodiation, the GCD curve doesn't change until the N/P ratio is tuned to 0.95. This could be due to that, it is difficult for the physical method to realize an accurately control of pre-sodiation degree.

Fig. R4 The galvanostatic charge curve of the hard carbon electrode.

This presumption could be further verified by the literature. Balbuena et al.^{R15} from the Pacific Northwest National Laboratory reported a lithium-pretreated HC to construct the full-cell, whereby no extra Na was pre-sodiated. However, as they regulated the N/P ratio to 1.5, their voltage profile of full-cell is very similar to that of Na₃V₂(PO₄)₃

in half-cell (Fig. R5).

Fig. R5 The galvanostatic charge/discharge curve of $\text{Na}_3\text{V}_2(\text{PO}_4)_3$ in **a** half-cell **b** full-cell collected from literature.^{R15}

Note that, the SEI formation in the first cycle could consume some Na, therefore, with higher pre-sodiation degree, more negative material (HCs) can be taken advantage. That is to say, increasing the pre-sodiation degree could be identical to improving the negative material ratio. On the other hand, the electrochemical pre-sodiation method allows a very accurate control on the pre-sodiation degree. We further conducted an experiment to evaluate the effect of the pre-sodiation degree on the curve shape with the N/P ratio unchanged. Sample EPS1-6 represents different pre-sodiation degree ($\text{EPS1} > \text{EPS2} > \text{EPS3} > \text{EPS4} > \text{EPS5} > \text{EPS6}$). It can be seen, with the same N/P ratio (~ 1.3), and the same pre-sodiation method, the GCD curve shape varies with the pre-sodiation degree (Fig. R6).

Fig. R6 The galvanostatic charge/discharge curve of full-cell with various pre-sodiated state of hard carbon.

An extreme case could also happen with too much Na pre-sodiated, i.e. the HCs doesn't participate in the full-cell function. In this way, the metal sodium could serve as the negative electrode, the full-cell will present the same GCD curve shape with the half-cell.

② Re-evaluation of the full-cell performance with smaller N/P ratio and electrochemical pre-sodiation method

Ideally, the as-equipped full-cell should make use of the capacity of both electrodes as much as possible, and meanwhile optimize the energy density and power density. In our revised manuscript, the electrochemical pre-sodiation method was applied to realize a well control of the pre-sodiation degree, and the N/P ratio was set as 1.2~1.4. The full-cell constructed in this way with hard carbon (HC), demonstrates a curvy plot below 3.2 V, which is different from that of half-cell. The as constructed full-cell delivered a high specific capacity of 60.4 mAh g⁻¹ in the voltage range of 2.3–4.14 V at 0.2 C at RT, corresponding to an energy density of 202 Wh kg⁻¹ (both the specific capacity and the energy density are calculated based on the total mass of cathode and anode). The full-cell also presents an outstanding rate performance up to 20 C (Fig. R7b-c & Fig. 6b-c), along with a power density of 7.75 kW kg⁻¹ (Fig. R7e & Fig. 6e). To explore the practical feasibility of the full battery under harsh conditions, the full cell was tested

at $-40\text{ }^{\circ}\text{C}$, which delivered a specific capacity of 57.7 mAh g^{-1} and 187.9 Wh kg^{-1} at 0.2 C (Fig. R7c & Fig. 6c), with a remarkably high-capacity retention of 95.5% of that at room temperature (Fig. R7e & Fig. 6e).

Fig. R7/6. Electrochemical performance of the full-cell. **a** Diagram of the P2-NaMNNb cathode//hard carbon anode full-cell. **b** The electrochemical curves of the full-cell with various current density. **c** Rate performance of the full-cell at $25\text{ }^{\circ}\text{C}$ and $-40\text{ }^{\circ}\text{C}$. **d** Cycling performance in a rate of 5 C at $25\text{ }^{\circ}\text{C}$ and 0.5 C at $-40\text{ }^{\circ}\text{C}$. **e** The comparison of the energy/power density with other reported literatures. ^{R8, R21-R22, R24}

An interesting phenomenon is that, the LT (low temperature) performance of the full-cell can be significantly affected by the N/P ratio. The full-cell constructed with $N/P < 1.5$ shows obvious decay of voltage platform under 3.2 V at $-40\text{ }^{\circ}\text{C}$, which could be improved by increasing the N/P ratio (Fig. R8). This could be mainly ascribed to the unsatisfying transport kinetics and obvious voltage polarization of hard carbon under LT. Therefore, it might be necessary to equip the full-cell with relatively high N/P ratio to offset the capacity decay of hard carbon, and sacrifice some RT (room temperature) performance. This is a complicated problem, which would be systematically studied in the future work.

Fig. R8 The galvanostatic charge/discharge curve of full-cell at -40 °C with different N/P ratio.

Table. R1. The comparison of full-cell performance with different pre-sodiation methods.

Cathode	Voltage range	Anode	Pre-sodiation method	Energy /Power density (Wh kg ⁻¹ /W kg ⁻¹)	Reference
Na _{0.67} Mn _{0.6} Ni _{0.2} Cu _{0.1} Co _{0.1} O ₂	1.9-4.2	Hard carbon	Electrochemical pre-sodiation	208/Not given	R1
Na _{2/3} Ni _{1/3} Mn _{2/3} O ₂ nanofibers	1.5-4.0	Hard carbon	Electrochemical pre-sodiation	212.2/Not given	R16
P2-layered Na _{0.7} Mg _{0.2} [Fe _{0.2} Mn _{0.6} □ _{0.2}]O ₂	1.5-4.5	Hard carbon	Electrochemical pre-sodiation	165/Not given	R17
Bipolar P2-Na _{0.6} [Cr _{0.6} Ti _{0.4}]O ₂	1.4-3.0	Na _{0.6} [Cr _{0.6} Ti _{0.4}]O ₂	Electrochemical pre-sodiation	94/Not given	R18
Na _{0.75} Mg _{0.08} Co _{0.10} Ni _{0.2} Mn _{0.6} O ₂	1.0-4.2	Hard carbon	Electrochemical pre-sodiation	309.7 (based on cathode material)/not given	R19
Na _{7/9} Cu _{2/9} Fe _{1/9} Mn _{2/3} O ₂	1.0-4.2	Hard carbon	Electrochemical pre-sodiation	195/Not given	R20
VOPO ₄	2-4.3	Na ₂ Ti ₃ O ₇	Electrochemical pre-sodiation	220/1600	R21
Na ₃ Fe ₂ (PO ₄) ₂ (P ₂ O ₇)	0-3.5	FBO@C	Electrochemical pre-sodiation	175/1680	R22
[Na _{0.67} Zn _{0.05}]Ni _{0.18} Cu _{0.1} Mn _{0.67} O ₂	2.4-4.25	Hard carbon	Direct contact to Na	217.9/2811	R9
P2-Na _{0.67} Ni _{0.23} Mg _{0.1} Mn _{0.67} O ₂	2.4-4.25	Hard carbon	Direct contact to Na	249.9/1700	R8
Na _{0.67} [Ni _{0.1} Fe _{0.1} Mn _{0.8}]O ₂	1.4-4.2	Hard carbon	Direct contact to Na	542(based on cathode material)/3900	R23
Na _{0.78} Ni _{0.31} Mn _{0.67} Nb _{0.02} O ₂	2.3-4.14	Hard carbon	Direct contact to Na	216/6600	Our work
Na _{0.78} Ni _{0.31} Mn _{0.67} Nb _{0.02} O ₂	2.3-4.14	Hard carbon	Electrochemical pre-sodiation	202/7747	Our work

Supplementary reference

- R1. Zhao, Q., et al. Tuning oxygen redox chemistry of P2-type manganese-based oxide cathode via dual Cu and Co substitution for sodium-ion batteries. *Energy Storage Mater.* **41**, 581-587 (2021).
- R2. Ding, F., et al. A Novel Ni-rich O3-Na[Ni_{0.60}Fe_{0.25}Mn_{0.15}]O₂ Cathode for Na-ion Batteries. *Energy Storage Mater.* **30**, 420-430 (2020).
- R3. Guo, Y.-J., et al. Boron-doped sodium layered oxide for reversible oxygen redox reaction in Na-ion battery cathodes. *Nat. Commun* **12**, 5267 (2021).
- R4. Shen, Q., et al. Transition-Metal Vacancy Manufacturing and Sodium-Site Doping Enable a High-Performance Layered Oxide Cathode through Cationic and Anionic Redox Chemistry. *Adv. Funct. Mater.* **31**, 2106923 (2021).
- R5. Xiao, Y., et al. A Stable Layered Oxide Cathode Material for High-Performance Sodium-Ion Battery. *Adv. Energy Mater.* **9**, 1803978 (2019).
- R6. Dewar, D. & Glushenkov, A. M. Optimisation of sodium-based energy storage cells using pre-sodiation: a perspective on the emerging field. *Energy Environ. Sci.* **14**, 1380-1401 (2021).
- R7. Wang, Y., et al. Pillar-beam structures prevent layered cathode materials from destructive phase transitions. *Nat. Commun* **12**, 13 (2021).
- R8. Peng, B., Sun, Z., Zhao, L., Li, J. & Zhang, G. Dual-Manipulation on P2-Na_{0.67}Ni_{0.33}Mn_{0.67}O₂ Layered Cathode toward Sodium-Ion Full Cell with Record Operating Voltage Beyond 3.5 V. *Energy Storage Mater.* **35**, 620-629 (2021).
- R9. Peng, B., et al. Unusual Site-Selective Doping in Layered Cathode Strengthens Electrostatic Cohesion of Alkali-Metal Layer for Practicable Sodium-Ion Full Cell. *Adv. Mater.* 2103210
- R10. Jo, J. H., et al. A new pre-sodiation additive for sodium-ion batteries. *Energy Storage Mater.* **32**, 281-289 (2020).
- R11. Niu, Y.-B., et al. High-Efficiency Cathode Sodium Compensation for Sodium-Ion Batteries. *Adv. Mater.* **32**, 2001419 (2020).
- R12. Zhang, B., et al. Insertion compounds and composites made by ball milling for advanced sodium-ion batteries. *Nat. Commun* **7**, 10308 (2016).
- R13. Ding, F., et al. Additive-Free Self-Presodiation Strategy for High-Performance Na-Ion Batteries. *Adv. Funct. Mater.* **31**, 2101475 (2021).
- R14. Park, Y. J., et al. A New Strategy to Build a High-Performance P'2-Type Cathode Material through Titanium Doping for Sodium-Ion Batteries. *Adv. Funct. Mater.* **29**, 1901912 (2019).
- R15. Xiao, B., et al. Lithium-Pretreated Hard Carbon as High-Performance Sodium-Ion Battery Anodes. *Adv. Energy Mater.* **8**, 1801441 (2018).
- R16. Liu, Y., et al. Hierarchical Engineering of Porous P2-Na_{2/3}Ni_{1/3}Mn_{2/3}O₂ Nanofibers Assembled by Nanoparticles Enables Superior Sodium-Ion Storage Cathodes. *Adv. Funct. Mater.* **30**, 1907837 (2020).
- R17. Li, X.-L., et al. Whole-Voltage-Range Oxygen Redox in P2-Layered Cathode Materials for Sodium-Ion Batteries. *Adv. Mater.* **33**, 2008194 (2021).
- R18. Wang, Y., Xiao, R., Hu, Y.-S., Avdeev, M. & Chen, L. P2-Na_{0.6}[Cr_{0.6}Ti_{0.4}]O₂ cation-disordered electrode for high-rate symmetric rechargeable sodium-ion batteries. *Nat. Commun* **6**, 6954 (2015).
- R19. Shi, Y., et al. Unlocking the potential of P3 structure for practical Sodium-ion batteries by fabricating zero strain framework for Na⁺ intercalation. *Energy Storage Mater.* **37**, 354-362 (2021).
- R20. Li, Y., et al. Air-Stable Copper-Based P2-Na_{7/9}Cu_{2/9}Fe_{1/9}Mn_{2/3}O₂ as a New Positive Electrode Material for Sodium-Ion Batteries. *Advanced Science* **2**, 1500031 (2015).
- R21. Li, H., et al. An advanced high-energy sodium ion full battery based on nanostructured Na₂Ti₃O₇/VOPO₄ layered materials. *Energy Environ. Sci.* **9**, 3399-3405 (2016).

-
- R22. Cao, Y., et al. All-Climate Iron-Based Sodium-Ion Full Cell for Energy Storage. *Adv. Funct. Mater.* **31**, 2102856 (2021).
- R23. Choi, J. U., Jo, J. H., Park, Y. J., Lee, K.-S. & Myung, S.-T. Mn-Rich P'2-Na_{0.67}[Ni_{0.1}Fe_{0.1}Mn_{0.8}]O₂ as High-Energy-Density and Long-Life Cathode Material for Sodium-Ion Batteries. *Adv. Energy Mater.* **10**, 2001346 (2020).
- R24. Fang, Y., et al. MXene-Derived Defect-Rich TiO₂@rGO as High-Rate Anodes for Full Na Ion Batteries and Capacitors. *Nano-Micro Letters* **12**, 128 (2020).

Reviewers' Comments:

Reviewer #2:

Remarks to the Author:

The energy density of full cells, 202 Wh/kg, was calculated based on the total mass of cathode and anode. It seems that the value was obtained based on the cathode material. See that the capacity obtained in the full cell is about 60 mAh/g. Does the total mass contain the weight of current collectors? The authors need to clarify it.

Different from the half-cell capacity (~ 90 mAh/g at 0.2C), the capacity measured in the full cell was about ~ 60 mAh/g at the same condition. The authors need to explain the possible reason for the reduction of capacity in the full cells.

Figure 6e. No explanation in the text. How did the authors obtained the energy density and power density, using half-cells or full cells data?

Even though I have re-reviewed this paper, it is very confusing what is the merit of the limited use of Ni^{2+/3+} redox couple that induces small capacity, compared to many other high capacity P2 type materials? The authors did not compare the corresponding energy density of the present materials with well-known high capacity P2 cathodes. Why? The authors need to compare the energy density plot with other P2 cathode materials to highlight the importance of this work. I agree with the electrode performance made at low temperature. However, the low capacity by the Ni^{2+/3+} redox at room temperature does not highlight the merit of the present material compared to high capacity P2 cathodes in terms of capacity and energy density.

Manuscript Number: NCOMMS-21-32268B-Z

Title: Regulating Surface and Local Chemistry in High Na-content P2-Cathode For Ultrahigh Power and Low-Temperature Sodium-Storage

Dear Editor and Reviewers,

Thank you very much for arduous work and valuable comments to review our manuscript. They are all valuable and very helpful for revising and improving our paper, as well as the important guiding significance to our work. We have carefully studied the comments and conducted more experiments to support our findings and below are our detailed, point-to-point responses.

Sincerely yours,

Yufeng Zhao

Comments from Reviewer#2:

The energy density of full cells, 202 Wh/kg, was calculated based on the total mass of cathode and anode. It seems that the value was obtained based on the cathode material. See that the capacity obtained in the full cell is about 60 mAh/g. Does the total mass contain the weight of current collectors? The authors need to clarify it.

Different from the half-cell capacity (~90 mAh/g at 0.2C), the capacity measured in the full cell was about ~60 mAh/g at the same condition. The authors need to explain the possible reason for the reduction of capacity in the full cells.

Figure 6e. No explanation in the text. How did the authors obtained the energy density and power density, using half-cells or full cells data?

Even though I have re-reviewed this paper, it is very confusing what is the merit of the limited use of Ni^{2+/3+} redox couple that induces small capacity, compared to many other high capacity P2 type materials? The authors did not compare the corresponding energy density of the present materials with well-known high capacity P2 cathodes. Why?

The authors need to compare the energy density plot with other P2 cathode materials to highlight the importance of

this work. I agree with the electrode performance made at low temperature. However, the low capacity by the Ni^{2+/3+} redox at room temperature does not highlight the merit of the present material compared to high capacity P2 cathodes in terms of capacity and energy density.

Response:

Q1. The energy density of full cells, 202 Wh/kg, was calculated based on the total mass of cathode and anode. It seems that the value was obtained based on the cathode material. See that the capacity obtained in the full cell is about 60 mAh/g. Does the total mass contain the weight of current collectors? The authors need to clarify it.

A1: Thanks for your valuable comments. In this work, both the specific capacity (60.4 mAh g⁻¹), the energy density (202 Wh kg⁻¹) and power density of full-cell are calculated based on the total mass of active material in cathode and anode, not including the weight of current collectors. Whereby, the specific capacity is calculated based on the equation^{r1} $C_t = \frac{I \times t}{m_c + m_a}$ (C_t (mAh g⁻¹) is the specific discharge capacity of the full-cell, I (mA) is the current density of the cell, t (h) is the total time of the discharge, m_c and m_a are the mass loading of P2-NaMNNb on the cathode and hard carbon on the anode, respectively), and the energy density of full-cell is calculated based on the equation^{r2} $E = C_t \times U_a$ (E represents the specific energy density (Wh kg⁻¹), U_a is the average discharge voltage of the cell). The power density of full-cell is calculated based on the equation^{r3} $P = \frac{E}{t}$ (P (W kg⁻¹) is the power density of the full-cell).

The corresponding corrections are made in the manuscript and supplementary information and highlighted in yellow.

Q2. Different from the half-cell capacity (~90 mAh/g at 0.2C), the capacity measured in the full cell was about ~60 mAh/g at the same condition. The authors need to explain the possible reason for the reduction of capacity in the full cells.

A2: Thank you for raising this question. It is a general phenomenon, that the specific capacity value of the full-cell (based on the total active mass of both electrodes) is smaller than that of the half-cell (based on the active mass of cathode only). For example, Zhang et al⁴ equipped the half-cell (cathode: $[\text{Na}_{0.67}\text{Zn}_{0.05}]\text{Ni}_{0.18}\text{Cu}_{0.1}\text{Mn}_{0.67}\text{O}_2$) with specific capacity of 103 mAh g⁻¹, while the specific capacity of full-cell is around 62 mAh g⁻¹; and Yu et al⁵ assembled the half-cell (cathode: $\text{Na}_3\text{V}_2(\text{PO}_4)_3/\text{C}$) with specific capacity of 113 mAh g⁻¹, while the specific capacity of full-cell is 60.5 mAh g⁻¹.

The following reasons can be considered to explain this phenomenon:

In a typical full-cell sodium ion battery, only the cathode material (in this case, P2-NaMNNb) can serve as the sodium source, if no extra Na⁺ is introduced. That means, the specific capacity of full-cell can only be obtained through the Na⁺ extracting from the cathode material. Theoretically, the total amount of Na⁺ that could be extracted from the cathode material are certain, but unlike the half-cell only the loading mass of cathode material is calculated, the capacity of full-cell (60.4 mAh g⁻¹) was obtained based on the total active mass of both electrodes.

Besides, the corresponding full-cell performance based only on the active mass of cathode materials is also calculated which is 86.7 mAh g⁻¹ @ 0.2 C. It is noticed that, this value is slightly lower than that obtained from half-cell (~96 mAh g⁻¹), which might be due to that a little bit of Na⁺ extracted from the cathode would be consumed in forming SEI on the anode side. Although the pre-sodiation can compensate the Na⁺ consumption in SEI, during the construction of the full-cell the SEI can be damaged in some degree, which would then lead to a slight decrease of the specific capacity (based on loading mass of the cathode material).

In some work, extra Na⁺ would be introduced by high-degreed pre-sodiation of the anode materials, which would increase the specific capacity of the full-cell accordingly. However, in our work, as the pre-sodiation

degree is strictly controlled, i.e. the pre-sodiated Na^+ ions are only to compensate the consumption of Na^+ in forming the irreversible SEI film. Therefore, the P2-NaMNNb serves as the only Na^+ source in the full-cell.

The corresponding explanations are added to the manuscript and highlighted in yellow.

Q3: Figure 6e. No explanation in the text. How did the authors obtained the energy density and power density, using half-cells or full cells data?

A3: Thank you for this question. The energy density and power density data in Fig.6e is based on the electrochemical performance of full-cell. The calculation methods for energy density (E) and power density (P) has been explained in the answer to Q1.

The corresponding revisions are made in the manuscript and highlighted in yellow.

Q4. Even though I have re-reviewed this paper, it is very confusing what is the merit of the limited use of $\text{Ni}^{2+/3+}$ redox couple that induces small capacity, compared to many other high capacity P2 type materials? The authors did not compare the corresponding energy density of the present materials with well-known high capacity P2 cathodes. Why? The authors need to compare the energy density plot with other P2 cathode materials to highlight the importance of this work. I agree with the electrode performance made at low temperature. However, the low capacity by the $\text{Ni}^{2+/3+}$ redox at room temperature does not highlight the merit of the present material compared to high capacity P2 cathodes in terms of capacity and energy density.

A4: Thanks for your positive comments about the low temperature performance of our material.

The merit of choosing such a relatively narrow voltage potential region can be considered form the following aspects:

(1) First of all, unlike LIBs, the target applications of sodium-ion batteries are the fields that don't require high energy density, but has special request for the low cost, high safety and stability as well as low temperature

performance, such as large-scale electricity grids, low speed HEV etc, attributed to the abundant and cheap Na sources, and smaller desolvation barrier of Na^+ .^{r6,r7}

(2) As for the P2-type layered oxides, they demonstrate higher theoretical capacity and higher rate performance than O3-type oxides. However, their high theoretical capacity can only be obtained in a wide potential window (1.5-4.4V).^{r8,r9} At the low voltage region (1.5-2.2 V), the P2-material will encounter serious Jahn-Teller effect caused by the redox of $\text{Mn}^{3+}/\text{Mn}^{4+}$, while irreversible phase transformation would happen at high voltage region (> 4.2 V), which restrains the practical application of P2-type cathode materials.^{r10,r11}

(3) On the other hand, expanding the cut off voltage to a lower voltage, would inevitably reduce the practical working potential of the full-cell and hence limit their practical applications in the fields requiring high voltage.

(4) It is reported that, the specific discharge capacity within the low voltage range of P2-type cathode often corresponds to the excess Na^+ insertion into the structure (the complete desodiation of the P2-type cathode can only happen at high voltage, which encounters irreversible phase transition above 4.2V) due to the low initial charge capacity deliver. Therefore to realize the high specific capacity and meanwhile keep the voltage below 4.2 V, it would require additional sodium sources, which would not applicable in large-scale applications.^{r12,r13}

For example, it needs about ~ 0.7 mol extra Na^+ to be desodiated from $\text{Na}_x\text{Mn}_{0.67}\text{Ni}_{0.33}\text{O}_2$ ($x \leq 0.67$) to achieve the 180 mAh g^{-1} within the 1.5-4.2 V.

Therefore, for the P2-type Na-Mn-Ni-O system, efficient utilization of the specific capacity within the medium voltage region ($2.2 \text{ V} < \text{Voltage} < 4.15 \text{ V}$) is of great importance for practical applications. However, a typical P2- $\text{Na}_x\text{Mn}_{0.67}\text{Ni}_{0.33}\text{O}_2$ ($x \leq 0.67$) can only present a low specific capacity of 80~85 mAh g^{-1} within this range (2.2-4.15 V) compensated by $\text{Ni}^{2+/3+}$ redox reaction.^{r14,r15} Recent works have shown that, increasing the Na-content in P2-oxides could improve the capacity in the medium voltage region.^{r16-r18} For example, Meng et

al.^{r16} reported a $\text{Na}_{0.78}\text{Ni}_{0.23}\text{Mn}_{0.69}\text{O}_2$ to stimulate redox reaction of $\text{Ni}^{2+}/\text{Ni}^{3+}/\text{Ni}^{4+}$ below 4.1 V, which achieved a specific capacity of $\sim 100 \text{ mAh g}^{-1}$ within 2.0~4.1 V. Hu et al.^{r17} reported $\text{Na}_{45/54}\text{Li}_{4/54}\text{Ni}_{16/54}\text{Mn}_{34/54}\text{O}_2$, the material could deliver the specific capacity of 103.4 mAh g^{-1} within 2.0~4.0 V with the redox reaction of $\text{Ni}^{2+}/\text{Ni}^{3+}/\text{Ni}^{4+}$. Some P2-type cathode with Na-Al-Fe-O or Na-Mn-Fe-O system (such as $\text{Na}_{0.67}\text{Al}_{0.1}\text{Mn}_{0.9}\text{O}_2$ ^{r19}, $\text{Na}_{0.67}\text{Mn}_{0.5}\text{Fe}_{0.5}\text{O}_2$ ^{r20}) can deliver 160-180 mAh g^{-1} with raised the redox of $\text{Mn}^{3+}/\text{Mn}^{4+}$ voltage ($\sim 2.5 \text{ V}$), but the average voltage is around 2.5 V and the rate performance is unsatisfying, as well as the poor cycling performance because of the Jahn-Teller effect induced by $\text{Mn}^{3+}/\text{Mn}^{4+}$.

In this work, we demonstrate a trace Nb doped $\text{Na}_{0.78}\text{Mn}_{0.67}\text{Ni}_{0.33}\text{O}_2$, which can also take advantage of the $\text{Ni}^{2+}/\text{Ni}^{3+}/\text{Ni}^{4+}$ redox reaction to show a specific discharge capacity of 96.4 mAh g^{-1} within 2.4~4.15 V. The trace Nb doping could simultaneously improve the Na^+ diffusion capability by modulating the local structure of P2- NaMNNb , and prevent the metal element dissolving by triggering the surface pre-construction.

Ex-situ X-ray absorption spectroscopy was performed again to confirm the $\text{Ni}^{2+}/\text{Ni}^{3+}/\text{Ni}^{4+}$ redox reaction happened upon charge/discharge (Fig. R1/Supplementary Fig. 13). It can be seen that the K-edge of Ni shifts to a higher energy region upon charge, indicating that Ni^{2+} is oxidized. The energy shift at 4.15 V is $\sim 2.4 \text{ eV}$, which is larger than 2 eV for $\text{Ni}^{2+}/\text{Ni}^{3+}$ redox reaction reported in literature^{r16,r21,r22}, suggesting that the Ni ions are responsible for charge compensation during the charge process and the valance state of Ni ions are oxidized from Ni^{2+} to the mixture of Ni^{3+} and Ni^{4+} (Fig. R1b/Supplementary Fig. 13b). It is estimated that the valance state of Ni ions is +3.2 at the end of charging process, which can deliver the specific capacity of 97.4 mAh g^{-1} , approximately approach the experimental results of 100 mAh g^{-1} at charging process. On the contrary, the energy shift of K-edge of Mn during electrochemical compensation is not obvious, indicating that Mn ion exhibit minor participation during the electrochemical reaction and remain +4 at the end of charging process (Fig. R1a/

Supplementary Fig. 13a). Therefore, Na⁺ diffusion capability of P2-NaMNNb has been greatly improved with the dual effects of bulk modulation and surface pre-construction, thus giving rise to redox reaction of Ni²⁺/Ni³⁺/Ni⁴⁺ with high voltage plateaus (3.2-3.7 V).

(5) Furthermore, the DFT calculation reveals that, the calculated Na⁺ diffusion barrier at Na deficient status, is much smaller than that at Na deficient state (0.22 vs. 0.667 eV) (Fig. R2b/Fig. 5i), indicating better Na⁺ mobility within the high voltage range (3.2-3.7 V) than low voltage range (1.5-2.2 V). Therefore, by choosing proper voltage range, we can maximize the rate capability, and hence improve the low temperature performance.

(6) In fact, our as prepared P2-NaMNNb also exhibits a high specific capacity of 182.1 mAh g⁻¹ in the voltage region of 1.5-4.15 V, which remains 88.8 mAh g⁻¹ at 30 C, which is superior to that of other cathode in the wide voltage range. The electrochemical performance of P2-NaMNNb in the potential window of 1.5-4.15 V, as well as the comparison with literatures are added to the supporting information (Fig. R3/Supplementary Fig. 15).

Considering all these aspects, we limited the voltage region between 2.4-4.15 V to sacrifice some capacity within low voltage range and make full use of the capacity of high voltage platform (3.2-3.7 V). To clarify this point, we have sorted out the electrochemical performance (average voltage, specific capacity and rate performance) of P2-type cathode material of half-cell, and (energy density/power density and the max current density) of full-cell reported by researchers and added in our revised manuscript with Fig. R4/Supplementary Fig. 16 and Table R1/Supplementary Table 5, as well as Table R2/Supplementary Table 6. Although the average voltage and specific capacity are not the highest among the reported materials, the rate performance of P2-NaMNNb in wide and narrow voltage region outperforms other cathode materials at room temperature.

The corresponding revisions are made in the manuscript and supplementary information and highlighted in yellow.

Fig. R1/Supplementary Fig. 13 The ex-situ XANES of P2-NaMNNb with different cutoff voltage a Mn ion, b Ni ion.

Fig. R2a-b/ Fig. 5h-i Calculated Na^+ ion diffusion pathways of P2-NaMNNb with (a) Na deficient status, (b) Na rich status, and the corresponding Na^+ migration energy barriers of P2-NaMN and P2-NaMNNb.

Fig. R3/Supplementary Fig. 15 a Charge–discharge curves of P2-NaMNNb at 25 °C in 1.5-4.15 V. b The corresponding rate performance at 25 °C.

Fig. R4/Supplementary Fig. 16 Comparison of half-cell performance with reported cathode materials for SIBs. Plots of specific capacity versus operating voltage with calculated energy density, as well as the maximum current density curves, a in narrow voltage range, b in wide voltage range. The data are consistent with those in Supplementary Table 5.

Table. R1/Supplementary Table 5 The comparison of electrochemical performance of various P2-Type cathode oxide material in half-cell.

	Cathode	Average voltage and potential window (V)	Specific capacity (mAh g⁻¹)	Rate performance and corresponding current density (mAh g⁻¹ / A g⁻¹)	Reference
Wide-voltage range	Na _{0.67} Al _{0.1} Mn _{0.9} O ₂	2.5 (2-4)	175	74 (4.8 A g ⁻¹)	r19
	Na _{2/3} Ni _{1/3} Mn _{2/3} O ₂ nanofibers	2.25 (1.5-4.0)	160	73.4 (3.5 A g ⁻¹)	r23
	NaPO ₃ -coated Na _{2/3} [Ni _{1/3} Mn _{2/3}]O ₂	3.2 (1.5-4.3)	194	115 (1.12 A g ⁻¹)	r24
	Na _{0.67} Mn _{0.5} Fe _{0.47} Al _{0.03} O ₂	2.6 (1.5-4.2)	167	67 (0.4 A g ⁻¹)	r25
	Na _{0.5} Ni _{0.1} Co _{0.15} Mn _{0.65} Mg _{0.1} O ₂	2.3 (1.5-4.0)	153.8	97.5 (0.8 A g ⁻¹)	r26
	Na _{0.67} [Mn _{0.61} Ni _{0.28} Sb _{0.11}]O ₂	2.25 (1.8-4.2)	140	82 (1.4 A g ⁻¹)	r27
	Na _{2/3} Zn _{1/4} Mn _{3/4} O ₂	2.63 (1.5-4.5)	202.4	140 (0.2 A g ⁻¹)	r28
	Na _{0.67} [Ni _{0.1} Fe _{0.1} Mn _{0.8}]O ₂	2.75 (1.5-4.3)	220	120 (1.3 A g ⁻¹)	r29
	Na _{2/3} Mn _{1/2} Co _{1/3} Ni _{1/6} O ₂	3.2 (1.5-4.5)	156	70 (0.5 A g ⁻¹)	r9
	Na _{0.8} Mn _{0.6} Co _{0.2} Mg _{0.2} O ₂	2.65 (1.6-4.4)	176	68.8 (1.36 A g ⁻¹)	r30
	Na_{0.78}Mn_{0.67}Ni_{0.31}Nb_{0.02}O₂	2.75 (1.5-4.15)	182.1	88.8 (5.52 A g⁻¹)	This work
Narrow-voltage range	Na _{0.78} Cu _{0.27} Zn _{0.06} Mn _{0.67} O ₂	3.6 (2.5-4.1)	88	73 (0.5 A g ⁻¹)	r31
	Na _{0.7} Mn _{0.6} Ni _{0.2} Mg _{0.2} O ₂	3.6 (2.5-4.2)	74	60 (4.3 A g ⁻¹)	r21
	Na _{2/3} Ni _{1/6} Mn _{2/3} Cu _{1/9} Mg _{1/18} O ₂	3.5 (2.5-4.15)	87.9	64.0 (3.6 A g ⁻¹)	r32
	Na _{2/3} Ni _{1/3} Mn _{2/3} O _{1.95} F _{0.05}	3.3 (2-4)	95.4	86.4 (1.7 A g ⁻¹)	r33
	Na _{0.85} Li _{0.12} Ni _{0.22} Mn _{0.66} O ₂	3.5 (2-4.3)	123	79.3 (4.2 A g ⁻¹)	r18
	Na _{0.75} Mg _{0.08} Co _{0.1} Ni _{0.2} Mn _{0.6} O ₂	3.32 (2-4.3)	123.8	79.8 (1.5 A g ⁻¹)	r34
	Na _{0.6} [Cr _{0.6} Ti _{0.4}]O ₂	3.5 2.5-3.85	74	61 (0.212 A g ⁻¹)	r35
	Na _{2/3} Ni _{1/3} Mn _{1/3} Ti _{1/3} O ₂	3.5 (2.5-4.15)	88	68.2 (3.46 A g ⁻¹)	r36

$[\text{Na}_{0.67}\text{Zn}_{0.05}]\text{Ni}_{0.18}\text{Cu}_{0.1}\text{Mn}_{0.67}\text{O}_2$	3.6 (2.5-4.35)	103	38 (3.4 A g ⁻¹)	r4
$\text{Na}_{0.78}\text{Mn}_{0.67}\text{Ni}_{0.31}\text{Nb}_{0.02}\text{O}_2$	3.4 (2.4-4.15)	96.4	65.8 (9.2 A g⁻¹)	This work

Table. R2/Supplementary Table 6 The comparison of electrochemical performance of various cathode material in full-cell.

Full-cell	potential window (V)	Energy density/power density (Wh kg ⁻¹ / W kg ⁻¹)	The max current density (A g ⁻¹)	Reference
Hard carbon [Na _{0.67} Zn _{0.05}]Ni _{0.18} Cu _{0.1} Mn _{0.67} O ₂	2.4-4.25	217.9	0.85	r4
Hard carbon Na _{2/3} Ni _{1/3} Mn _{2/3} O ₂ nanofibers	1.2-3.8	212.5	1.73	r23
Hard carbon P2/P3-Na _{0.7} Li _{0.06} Mg _{0.06} Ni _{0.22} Mn _{0.67} O ₂	2-4.2	218	0.595	r37
Hard carbon Na _{0.7} Mg _{0.2} [Fe _{0.2} Mn _{0.6□_x}]O ₂	1.4-4.4	165	Not given	r10
Hard carbon Na _{0.76} Ca _{0.05} [Ni _{0.23□_{0.08}} Mn _{0.69}]O ₂	1.9-4.2	257.6	2.4	r38
Hard carbon Na _{0.67} [Ni _{0.1} Fe _{0.1} Mn _{0.8}]O ₂	1.4-4.2	542 (based on cathode material)/3900	1.3	r29
Hard carbon Na _{0.67} Mn _{0.6} Ni _{0.2} Cu _{0.1} Co _{0.1} O ₂	1.9-4.2	208	2	r27
VOPO ₄ Na ₂ Ti ₃ O ₇	2-4.3	220/1600	0.5	r39
FBO@C Na ₃ Fe ₂ (PO ₄) ₂ (P ₂ O ₇)	0-3.5	175/1680	Not given	r1
Hard carbon Na _{0.67} Ni _{0.23} Mg _{0.1} Mn _{0.67} O ₂	2.4-4.25	249.9/1700	0.85	r40
Na_{0.78}Mn_{0.67}Ni_{0.31}Nb_{0.02}O₂	2.3-4.14	202/7747	3.68	This work

Supplementary reference

- r1. Cao, Y., et al. All-Climate Iron-Based Sodium-Ion Full Cell for Energy Storage. *Adv. Funct. Mater.* **31**, 2102856 (2021).
- r2. Linden, D. Handbook of batteries. *Fuel and Energy Abstracts* **36**, 265 (1995).
- r3. Fang, Y., et al. MXene-Derived Defect-Rich TiO₂@rGO as High-Rate Anodes for Full Na Ion Batteries and Capacitors. *Nano-Micro Letters* **12**, 128 (2020).
- r4. Peng, B., et al. Unusual Site-Selective Doping in Layered Cathode Strengthens Electrostatic Cohesion of Alkali-Metal Layer for Practicable Sodium-Ion Full Cell. *Adv. Mater.* **34**, 2103210 (2022).
- r5. Rui, X., et al. A Low-Temperature Sodium-Ion Full Battery: Superb Kinetics and Cycling Stability. *Adv. Funct. Mater.* **31**, 2009458 (2021).
- r6. Usiskin, R., et al. Fundamentals, status and promise of sodium-based batteries. *Nature Reviews Materials* **6**, 1020-1035 (2021).
- r7. Kubota, K., Kumakura, S., Yoda, Y., Kuroki, K. & Komaba, S. Electrochemistry and Solid-State Chemistry of NaMeO₂ (Me = 3d Transition Metals). *Adv. Energy Mater.* **8**, 1703415 (2018).
- r8. Xiao, Y., et al. A Layered-Tunnel Intergrowth Structure for High-Performance Sodium-Ion Oxide Cathode. *Adv. Energy Mater.* **8**, 1800492 (2018).
- r9. Liu, Z., et al. Ultralow Volume Change of P2-Type Layered Oxide Cathode for Na-Ion Batteries with Controlled Phase Transition by Regulating Distribution of Na⁺. *Angew. Chem. Int. Ed.* **60**, 20960-20969 (2021).
- r10. Li, X.-L., et al. Whole-Voltage-Range Oxygen Redox in P2-Layered Cathode Materials for Sodium-Ion Batteries. *Adv. Mater.* **33**, 2008194 (2021).
- r11. Wang, K., et al. Dopant Segregation Boosting High-Voltage Cyclability of Layered Cathode for Sodium Ion Batteries. *Adv. Mater.* **31**, 1904816 (2019).
- r12. Wang, H., et al. Electrochemical properties of P2-Na_{2/3}[Ni_{1/3}Mn_{2/3}]O₂ cathode material for sodium ion batteries when cycled in different voltage ranges. *Electrochim. Acta* **113**, 200-204 (2013).
- r13. Yuan, D., et al. P2-type Na_{0.67}Mn_{0.65}Fe_{0.2}Ni_{0.15}O₂ Cathode Material with High-capacity for Sodium-ion Battery. *Electrochim. Acta* **116**, 300-305 (2014).
- r14. Wang, P.-F., et al. Suppressing the P2-O2 Phase Transition of Na_{0.67}Mn_{0.67}Ni_{0.33}O₂ by Magnesium Substitution for Improved Sodium-Ion Batteries. *Angew. Chem. Int. Ed.* **55**, 7445-7449 (2016).
- r15. Singh, G., et al. High Voltage Mg-Doped Na_{0.67}Ni_{0.3-x}Mg_xMn_{0.7}O₂ (x = 0.05, 0.1) Na-Ion Cathodes with Enhanced Stability and Rate Capability. *Chem. Mater.* **28**, 5087-5094 (2016).
- r16. Ma, C., et al. Exploring Oxygen Activity in the High Energy P2-Type Na_{0.78}Ni_{0.23}Mn_{0.69}O₂ Cathode Material for Na-Ion Batteries. *J. Am. Chem. Soc.* **139**, 4835-4845 (2017).
- r17. Zhao, C., et al. Revealing High Na-Content P2-Type Layered Oxides as Advanced Sodium-Ion Cathodes. *J. Am. Chem. Soc.* **142**, 5742-5750 (2020).
- r18. Jin, T., et al. Realizing Complete Solid-Solution Reaction in High Sodium Content P2-Type Cathode for High-Performance Sodium-Ion Batteries. *Angew. Chem. Int. Ed.* **59**, 14511-14516 (2020).
- r19. Liu, X., et al. P2-Na_{0.67}Al_xMn_{1-x}O₂: Cost-Effective, Stable and High-Rate Sodium Electrodes by Suppressing Phase Transitions and Enhancing Sodium Cation Mobility. *Angew. Chem. Int. Ed.* **58**, 18086-18095 (2019).
- r20. Kumar, V. K., Ghosh, S., Biswas, S. & Martha, S. K. P2-Type Na_{0.67}Mn_{0.5}Fe_{0.5}O₂ Synthesized by Solution Combustion Method as an Efficient Cathode Material for Sodium-Ion Batteries. *J. Electrochem. Soc.* **168**, 030512 (2021).
- r21. Wang, Q.-C., et al. Tuning P2-Structured Cathode Material by Na-Site Mg Substitution for Na-Ion Batteries. *J. Am. Chem. Soc.* **141**, 840-848 (2019).

-
- r22. Yabuuchi, N., Yoshii, K., Myung, S.-T., Nakai, I. & Komaba, S. Detailed Studies of a High-Capacity Electrode Material for Rechargeable Batteries, $\text{Li}_2\text{MnO}_3\text{-LiCo}_{1/3}\text{Ni}_{1/3}\text{Mn}_{1/3}\text{O}_2$. *J. Am. Chem. Soc.* **133**, 4404-4419 (2011).
- r23. Liu, Y., et al. Hierarchical Engineering of Porous $\text{P2-Na}_{2/3}\text{Ni}_{1/3}\text{Mn}_{2/3}\text{O}_2$ Nanofibers Assembled by Nanoparticles Enables Superior Sodium-Ion Storage Cathodes. *Adv. Funct. Mater.* **30**, 1907837 (2020).
- r24. Jo, J. H., et al. Sodium-Ion Batteries: Building Effective Layered Cathode Materials with Long-Term Cycling by Modifying the Surface via Sodium Phosphate. *Adv. Funct. Mater.* **28**, 1705968 (2018).
- r25. Wang, H., et al. Different Effects of Al Substitution for Mn or Fe on the Structure and Electrochemical Properties of $\text{Na}_{0.67}\text{Mn}_{0.5}\text{Fe}_{0.5}\text{O}_2$ as a Sodium Ion Battery Cathode Material. *Inorg. Chem.* **57**, 5249-5257 (2018).
- r26. Zhu, Y.-F., et al. Manipulating Layered P2@P3 Integrated Spinel Structure Evolution for High-Performance Sodium-Ion Batteries. *Angew. Chem. Int. Ed.* **59**, 9299-9304 (2020).
- r27. Wang, Q.-C., et al. Tuning Sodium Occupancy Sites in P2-Layered Cathode Material for Enhancing Electrochemical Performance. *Adv. Energy Mater.* **11**, 2003455 (2021).
- r28. Wang, Y., et al. Ultralow-Strain Zn-Substituted Layered Oxide Cathode with Suppressed P2–O2 Transition for Stable Sodium Ion Storage. *Adv. Funct. Mater.* **30**, 1910327 (2020).
- r29. Choi, J. U., Jo, J. H., Park, Y. J., Lee, K.-S. & Myung, S.-T. Mn-Rich $\text{P'2-Na}_{0.67}[\text{Ni}_{0.1}\text{Fe}_{0.1}\text{Mn}_{0.8}]\text{O}_2$ as High-Energy-Density and Long-Life Cathode Material for Sodium-Ion Batteries. *Adv. Energy Mater.* **10**, 2001346 (2020).
- r30. Li, X.-L., et al. Stabilizing Transition Metal Vacancy Induced Oxygen Redox by $\text{Co}^{2+}/\text{Co}^{3+}$ Redox and Sodium-Site Doping for Layered Cathode Materials. *Angew. Chem. Int. Ed.* **60**, 22026-22034 (2021).
- r31. Yan, Z., et al. A Hydrostable Cathode Material Based on the Layered P2@P3 Composite that Shows Redox Behavior for Copper in High-Rate and Long-Cycling Sodium-Ion Batteries. *Angew. Chem. Int. Ed.* **58**, 1412-1416 (2019).
- r32. Xiao, Y., et al. A Stable Layered Oxide Cathode Material for High-Performance Sodium-Ion Battery. *Adv. Energy Mater.* **9**, 1803978 (2019).
- r33. Liu, K., et al. Insights into the Enhanced Cycle and Rate Performances of the F-Substituted P2-Type Oxide Cathodes for Sodium-Ion Batteries. *Adv. Energy Mater.* **10**, 2000135 (2020).
- r34. Shi, Y., et al. Unlocking the potential of P3 structure for practical Sodium-ion batteries by fabricating zero strain framework for Na^+ intercalation. *Energy Storage Mater.* **37**, 354-362 (2021).
- r35. Wang, Y., Xiao, R., Hu, Y.-S., Avdeev, M. & Chen, L. $\text{P2-Na}_{0.6}[\text{Cr}_{0.6}\text{Ti}_{0.4}]\text{O}_2$ cation-disordered electrode for high-rate symmetric rechargeable sodium-ion batteries. *Nat. Commun* **6**, 6954 (2015).
- r36. Wang, P.-F., et al. Na^+ /vacancy disordering promises high-rate Na-ion batteries. *Science Advances* **4**, eaar6018 (2018).
- r37. Zhou, Y.-N., et al. A P2/P3 composite layered cathode for high-performance Na-ion full batteries. *Nano Energy* **55**, 143-150 (2019).
- r38. Shen, Q., et al. Transition-Metal Vacancy Manufacturing and Sodium-Site Doping Enable a High-Performance Layered Oxide Cathode through Cationic and Anionic Redox Chemistry. *Adv. Funct. Mater.* **31**, 2106923 (2021).
- r39. Li, H., et al. An advanced high-energy sodium ion full battery based on nanostructured $\text{Na}_2\text{Ti}_3\text{O}_7/\text{VOPO}_4$ layered materials. *Energy Environ. Sci.* **9**, 3399-3405 (2016).
- r40. Peng, B., Sun, Z., Zhao, L., Li, J. & Zhang, G. Dual-Manipulation on $\text{P2-Na}_{0.67}\text{Ni}_{0.33}\text{Mn}_{0.67}\text{O}_2$ Layered Cathode toward Sodium-Ion Full Cell with Record Operating Voltage Beyond 3.5 V. *Energy Storage Mater.* **35**, 620-629 (2021).

Reviewers' Comments:

Reviewer #2:

Remarks to the Author:

Now, it is better than the previous versions. I recommend this work to be published in this leading journal.

Manuscript Number: NCOMMS-21-32268C

Title: Niobium-doped layered cathode material for high-power and low-temperature sodium-ion batteries.

Comments from Reviewer#2:

Reviewer #2 (Remarks to the Author):

Now, it is better than the previous versions. I recommend this work to be published in this leading journal.

Response:

Thank you very much for your positive comment and your recommendation on this article. We also highly appreciate your constructive and valuable advices and suggestions.